

**Upper-air secondary pollutants downward invade to planetary boundary layer by strong turbulence at the eastern steep slope of Tibetan Plateau: results from BLMP-SCB**

Suping Zhao [1,2], Shaofeng Qi [1,3], Jianjun He [4], Ye Yu [1,2], Longxiang Dong [1,2], Tong Zhang [1,2], Guo Zhao [1,2], Yiting Lv [1,3]

1. Key Laboratory of Cryospheric Science and Frozen Soil Engineering, Northwest Institute of Eco-Environment and Resources, Chinese Academy of Sciences, Lanzhou
730000, China.
2. Pingliang Land Surface Process & Severe Weather Research Station, Pingliang 744015, China.
3. University of Chinese Academy of Sciences, Beijing 100049, China.
4. State Key Laboratory of Severe Weather & Key Laboratory of Atmospheric
Chemistry of CMA, Chinese Academy of Meteorological Sciences, Beijing 100081, China.

Corresponding author: Suping Zhao (zhaosp@lzb.ac.cn)




**Abstract**

The comprehensive filed campaign is essential to deeply understand the interactions between aerosol and planetary boundary layer (PBL) meteorology, and implement of the relevant campaign is difficult, and thus data is scarce at the complex terrain. The first planetary boundary layer meteorology and pollution at western SiChuan Basin (BLMP-SCB) was conducted from December 2018 to January 2019. The campaign provides good chance for revealing the poorly-known the impact of PBL turbulence on profiles of air pollutants. The primary particulate matter (PM) pollutants rapidly decline with the increasing altitude, while the secondary ultraviolet PM ($UVPM_{sec}$) reduces more slowly and even shows a peak at 1.5–2.0 km above sea level. The regional and long-range transports are comparable between the primary and secondary PM pollutants during the campaign. The more uniform $UVPM_{sec}$ profiles during the nighttime are mainly modulated by thermodynamic (temperature) processes, while the secondary pollutants at PBL top downward invade into PBL by some dynamic processes, i.e., mechanical turbulence and wind shear, lending to more $UVPM_{sec}$ within PBL during the daytime. This study emphasizes the importance of turbulence and wind shear for the vertical profiles of air pollutants at complex terrain, especially at the sloped terrain. The results are helpful for understanding formation mechanism of heavy air pollution at the complex terrain, and then taking the targeted countermeasures.

**Keywords:** Planetary boundary layer; turbulence; air pollution; complex terrain; Tibetan Plateau





## 1 Introduction


Aerosol-planetary boundary layer (PBL) interactions are found to be one of the most important mechanisms deteriorating urban air quality near the ground surface (Li et al., 2021). The effect is largely dependent on chemical components of aerosol particles and their vertical distributions (Sun et al., 2025). Black carbon (BC) induces

heating in the upper PBL, and the resulting decreased surface heat flux substantially depresses the development of PBL and consequently enhances the occurrences of extreme haze pollution episodes, and this process is defined as the dome effect of BC (Ding et al., 2016). BC can also enhance PBL development depending on the properties and altitude of the BC layer. Slater et al. (2022) applied a high-resolution

model to quantify the impact of BC at different altitudes on PBL dynamics in Beijing, and found that BC within the PBL increases maximum PBL height by 0.4% but that the same loading of BC above the PBL can suppress PBL height by 6.5%. Briefly, the different optical properties of aerosol particles (absorption and scattering) at the varying altitudes exist contrasting impact on PBL dynamics, i.e., the stove, dome, and

umbrella effects of aerosol particles (Ma et al., 2020).

The vertical profiles of both chemical composition and the corresponding optical properties of aerosol particles are mainly influenced by vertical mixing and regional transport (Guan et al., 2024; Tian et al., 2017). The aerosol particles within PBL are

more modulated by vertical mixing of local emissions, while those above PBL are mainly affected by regional transport from upstream sources, such as biomass burning or dust (Yin et al., 2020; Zhao et al., 2019a). The biomass burning and coal combustion with the open-hearth furnace at mountaintop over Tibetan Plateau (TP) can be transported to above downstream basin due to mountain-valley breeze or

mountain-plain winds induced by terrain forcing (Zhao et al., 2023). Unlike BC particles, light-absorbing efficiencies of brown carbon (BrC) were found to be significantly increased with elevation from western Sichuan Basin to eastern TP, attributing to the enhancement in secondary formation and changing sources with the increasing elevation (Qi et al., 2025; Zhao et al., 2022). Dark BrC from biomass





burning contributes a substantial radiative effect of +0.208 W m$^{-2}$ (+0.02 to 0.68 W m$^{-2}$) via its solar radiation absorption, comparable to BC and far exceeding traditional BrC estimates (Wang et al., 2025). The strong light-absorbing aerosols over TP are transported to above basin and then deteriorates air quality within basin by aerosol-PBL feedbacks. Therefore, the long-range transport and aerosol-PBL feedback may

interact rather than act as two isolated processes as traditionally thought by investigating typical regional haze events over China (Huang et al., 2020).

Compared with the regional transport, the impact of vertical mixing on air pollution is poorly understood. The vertical mixing of energy, water vapor and pollutants easily

occurs around the large-scale terrain due to thermodynamic forcing. TP is considered as an important channel transporting Asian surface pollutants to the global stratosphere in response to strong "heat pump" (Bian et al., 2020; C. F. Zhao et al., 2020), which may be enhanced due to more rapid climate warming at the high-altitude regions (S. Y. Zhao et al., 2020). BC aerosols originating from South Asia can climb

over Himalaya mountains to reach inland TP and accelerate glacier melting by absorbing solar radiation (Kang et al., 2019). A dust aerosol layer was found at a height of 3–4 km above the ground at the northern slopes of TP, i.e., "suspended dust" (He et al., 2024), which closely related to strong turbulence and heatwave of desert underlying surface (Liu et al., 2024; Zhang et al., 2024). The heat contribution of dust

to the anomalously warm atmospheric layer over the Tarim Basin in spring and summer are 13.77% and 10.25% respectively, which seems a northward extension of TP heat source (Zhou et al., 2022). At eastern slopes of TP, the primary pollutants from Sichuan Basin can be transported to the areas with an altitude below 3.0 km (Yin et al., 2020).


Besides upward transports, TP is a global hotspot of stratospheric intrusion. The stratospheric intrusion was considered as a dominating factor of tropospheric ozone over the TP (Yang et al., 2022), especially in the areas with high surface ozone concentrations during their peak-value month (Yin et al., 2024). In the recent study of



Zhang et al. (2025), ozone surges within SiChuan Basin (SCB) were found to be jointly triggered by deep stratospheric intrusions and the Tibetan Plateau (TP) topographic forcing. The intruded $O_3$ over TP was transported into the downstream SCB by strong downdrafts along the TP's leeward slope. However, the studies are lack of in-situ observational evidence and only focus on ozone. The invasion of upper-air

secondary pollutants to PBL over deep basin is less studied and the relevant mechanisms are poorly known. Therefore, we conducted the first Boundary Layer Meteorology and Pollution at SiChuan Basin (BLMP-SCB) during the winter in 2018 (Zhao et al., 2023), which provides a good opportunity for deeply understanding the downward transport of secondary air pollutants. The results are useful for

understanding the change in air pollutants and then taking targeted measures.

## 2 Data and methods

### 2.1 Data from BLMP-SCB

  The first field campaign of Boundary Layer Meteorology and Pollution at

SiChuan Basin (BLMP-SCB) was conducted at a rural site (Sanbacun, 103°40′38″ E, 30°54′59″ N) of eastern foothills of Tibetan Plateau in winter of 2018, lasting about 40 days (Fig. 1). A tethered balloon was used to in-situ observe the vertical profiles of key $PM_1$ (mass and the carbonaceous components) and gaseous pollutants (CO, NO, $NO_2$, $O_3$, TVOC) and meteorological variables (temperature, RH) within PBL. The

variables were observed every three-hours (02:00, 05:00, 08:00, 11:00, 14:00, 17:00, 20:00, 23:00) during the campaign, which is helpful for understanding the PBL turbulence and its impacts. A lightweight low-cost multi-pollutant sensor package, developed by Pang et al. (2021), is very portable and suitable for aerial measurements, and thus it was carried by the tethered balloon during the campaign. The system

consists of electrochemical sensors measuring gaseous pollutants and an optical counter (OPC) for $PM_1$ with time resolution of 10 seconds. The performances of the sensors were verified by comparing with on-ground reference instruments (Pang et al., 2021), and it was substantiated to be a reliable device for aerial measurements of PM and gaseous pollutants within PBL.



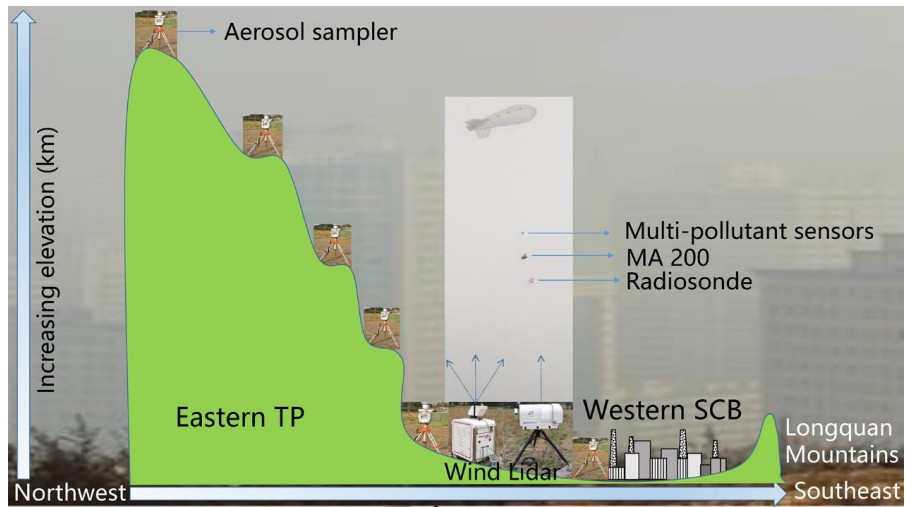


Fig. 1 Experimental set-up of 3D pollution and meteorology within PBL at the eastern foothills of Tibetan Plateau (TP).

The micorAeth MA200 (AethLabs, USA) was used to measure the mass

concentrations of carbonaceous components of aerosol particles at five wavebands (375 nm, 470 nm, 528 nm, 625 nm and 880 nm). The carbonaceous particles measuring at 880 nm by the instrument were usually interpreted as Black Carbon (BC), while those at 375 nm ultraviolet band were thought to be Ultraviolet Particulate Matter (UVPM). The MA200 draws an air sample at a flow rate of 100 ml

$\text{min}^{-1}$ through a 3 mm diameter portion of the filter media. The light attenuation (ATN) in response to absorbance of particles collected on the 'Sensing' spot is measured relative to an adjacent 'Reference' portion of the filter where no particles are accumulated. The temporal resolution of 5 seconds was set to match the other observations during the campaign.


A portable GPS upper-air sounding system (IMET-3050) was deployed to monitor the vertical profiles of temperature and RH by carrying the radiosonde (IMET-1-AB) on the tethered balloon (Li et al., 2017). The radiosonde has been widely used and validated (Haman et al., 2012), and there is a very slight difference



with the other radiosondes such as Vaisala RS92 (Trapp et al., 2016). The radiosonde

can be recycled during the campaign, which was verified by Zhao et al. (2023). The

temporal resolution of temperature and RH data ranged from 1 to 3 seconds for each

profile. The uncertainty of temperature and RH measurements was ±0.3 °C and ±5%

given by the manufacturer. The profiles of horizontal winds (speed and direction) and

vertical velocity were observed by a Doppler Wind Lidar (Windcube 200s, Leosphere,

France). The Lidar emits a fixed pulse signal into the atmosphere, and the frequency

of the pulse signal changes when the electromagnetic waves encounter the moving

particles, and thus the obtained data quality was closely related to aerosol particles.

The radial wind speed and direction can be retrieved by analyzing the frequency shift

from emitted to backscattered signals. The Lidar has four scanning modes for

different applications: plan position indicator (PPI), range height indicator (RHI),

line-of-sight (LOS), and Doppler beam swinging (DBS) scan modes (Lundquist et al.,

2017). The DBS mode was used in this campaign. The temporal resolutions of the

horizontal and vertical winds are 0.2 and 0.05 Hz, and the vertical resolution of 50 m

was used in this study. The wind data with signal-to-noise ratio lower than -26 dB

were removed from the raw data. The data were averaged hourly to combine with the

other measurements on the tethered balloon for understanding pollution-PBL

interactions. The wind data measured by the Lidar were validated by comparing with

the in-situ wind observations by Beijing 325-m meteorological tower, and no

significant difference was found between the Lidar and sonic wind anemometer for

wind speed and direction (Dai et al., 2020).

**2.2 Identification of potential source regions for BC and UVPM**

In regional and even the larger scales, we used HYbrid Single-Particle

Lagrangian Integrated Trajectory (HYSPLIT) model to determine the origin of air

masses and understand the difference of potential source regions between BC and

UVPM during the campaign. The 96-h backward trajectories arriving at the three

heights above ground level (100 m, 700 m and 1300 m, AGL) and initializing at the

hour of day, the same as the launch of tethered balloon, was calculated with Global



Data Assimilation System (GDAS) data (0.25°×0.25°) from National Centers for

Environmental Prediction (NCEP). Based on the trajectories, the concentration-weight

trajectory (CWT) method (Hsu et al., 2003) was used to determine the potential

source region of BC and UVPM. Additionally, combining BC and UVPM

concentrations obtained by MA200 on the balloon with winds measured by the Lidar,

the pollution roses were calculated at the heights of 100 m, 700 m and 1300 m AGL

by the Openair package of Rplot software. The difference of potential source regions

between BC and UVPM at regional and local scales can be more comprehensively

understood by combining CWT method with pollution rose.

**2.3 Cluster analysis methods**

Clustering analysis was widely used in big-data analysis of environmental field

(Sabaliauskas et al., 2013; Tunved et al., 2004), which has been considered to be a

preferred technique for extracting some more valuable information. The K-means

clustering technique splits the multi-dimensional data into pre-defined number of

subgroups, and clusters are as different as possible from each other, but as

homogeneous as possible within themselves, by iteratively minimizing the sum of

squared Euclidean distances from each member to its cluster centroid. Clustering

analysis was used to divide the UVPM profiles during the campaign into three groups

with comparable vertical structure of UVPM within groups. The K-means clustering

algorithm available in MATLAB© was used in this study.

**2.4 Post-processing of on-line measured carbonaceous aerosols by MA200**

The micorAeth MA200 may produce negative values in the lower mass

concentrations and the higher temporal resolution, contributing up to 30% of the

uncertainty for filter-based optical attenuation technique (Hagler et al., 2011), and

thus the obtained raw data for vertical profiles must be corrected before analyzing the

characteristics, especially for the in-situ observations at high-altitude. The optical

noise-reduction averaging (ONA) program was used to post-process the negative

values from our real-time profile measurements. The algorithm is to conduct variable





230 time-averaging of carbonaceous components measured by MA200 to reduce noise in
the data. The ONA algorithm results in significant noise reductions and much more
reasonable temporal changes in mass concentrations of carbonaceous particles (Cheng
and Lin 2013; Park et al., 2010).

235   The estimation of secondary organic carbon ($UVPM_{sec}$) is important for
determining the fraction of $UVPM_{sec}$ in UVPM. Initially, the ($UVPM_{sec}$ /BC) $_{min}$ ratio
was used as a proxy for the $UVPM_{pri}$ / UVPM ratio to estimate $UVPM_{sec}$ mass
concentrations (Castro et al., 1999). However, many studies have found that (UVPM
/BC) $_{min}$ ratio exhibits a degree of randomness in the actual observations, leading to

240 significant errors, especially for the low BC concentrations at the high-altitude
regions. To address this, Lim and Turpin (2002) have proposed sorting the UVPM/BC
ratios in ascending order and replacing the $UVPM_{pri}$ /BC ratio with the average value
of the top 10%–20% of the data, but it is lack of a universally applicable criterion for
determining the appropriate percentile range. In view of the distinct sources between

245 $UVPM_{sec}$ and BC, Millet et al. (2005) proposed a method estimating $UVPM_{sec}$
concentrations with the minimum correlation coefficient between UVPM and BC.
This approach is to identify the UVPM /BC ratio (denoted as (UVPM /BC) $_{pri}$) at
which the correlation between $UVPM_{sec}$ and BC is the weakest, and this ratio is used
as the $UVPM_{pri}$ /BC ratio. Following this method, Wu and Yu (2016) developed a

250 toolkit in Igor Pro for calculating $UVPM_{sec}$ mass concentration, significantly
enhancing the accuracy of $UVPM_{sec}$ estimation, as shown in Eqs. (1) and (2).

$$UVPM_{pri} = (UVPM/BC)_{pri} \times BC, \qquad (1)$$

$$UVPM_{sec} = UVPM - UVPM_{pri}. \qquad (2)$$

In Eq. (1), (UVPM/BC) $_{pri}$ represents the ratio of $UVPM_{pri}$ to BC concentrations

255 during the campaign.

**2.5 Calculation of mechanical turbulence and wind shear**

   PBL turbulence is an important mechanism for modulating exchanges of energy,
water vapor and greenhouse gases between land and atmosphere. The atmosphere is





heated by longwave radiation from land surface, and thus the thermal turbulence

mainly reflects the impact of land on atmosphere (Sun et al., 2006). In addition, the

mechanical turbulence, mostly induced by wind speed or directional shear, generally

represents the influence of the atmosphere on the land due to the increased wind speed

with altitude (Zhao et al., 2023). The mechanical turbulence index ($V_{TKE}$, unit: m s$^{-1}$)

was calculated with the below equation:

$$V_{TKE} = \sqrt{\tfrac{1}{2}\left(\overline{u'^2} + \overline{v'^2} + \overline{w'^2}\right)}, \qquad (3)$$

where, $u'$, $v'$ and $w'$ are the fluctuations of three-dimensional components of winds ($u$,

$v$ and $w$) during the campaign. The vertical profiles of the index can be obtained by

the above equation. The higher the index, the stronger the mechanical turbulence.

Combination the profiles of $V_{TKE}$ with air pollutants can be used to better understand

the downward transport of air pollutants at the eastern foothills of Tibetan Plateau.

Referring to Mahrt (2017), we defined several measures of the wind shear. The wind

speed shear was defined as

$$Sh \equiv |\delta V(\overline{u}, \overline{v})|, \qquad (4)$$

where $\delta V$ refers to differences of wind speed ($V$) between the adjacent measurement

levels. Additionally, $S_{vec}$ was defined as the magnitude of the vector shear based on

the vertical differences of the wind-speed components

$$S_{vec} \equiv \sqrt{(\delta \overline{u})^2 + (\delta \overline{v})^2}. \qquad (5)$$

The wind-directional shear can be quantified as the difference between the magnitude

of the vector shear and the speed shear

$$S_{dir} \equiv S_{vec} - Sh, \qquad (6)$$

where $S_{dir}$ is expressed in m s$^{-1}$.

## 3   Results and discussion

### 3.1 Vertical profiles of air pollutants

The aerosol-PBL meteorology feedbacks are very sensitive to the altitude of

carbonaceous components (Wang et al., 2018). In view of the increase in solar UV




radiation with altitude (Blumthaler et al., 1997), the slower decrease in UVPM than

BC with altitude leads to more significant impact of UVPM on PBL meteorology

(Zhao et al., 2023). To better reveal the mechanisms of more uniform UVPM profiles,

we further separated $UVPM_{sec}$ from UVPM with Eqs. (1–2) and calculated the ratio of

$UVPM_{sec}$ to UVPM (Fig. 2). As shown in Fig. 2, the primary UVPM ($UVPM_{pri}$)

rapidly decreased with altitude and were mainly trapped in the regions with the

altitude below 1.0 km, which was similar with BC profiles. The spikes of primary

carbonaceous components ($UVPM_{pri}$ and BC) at 02:00 at the altitudes ranging from

0.8 km to ~1.0 km were jointly induced by both regional transport and low PBL

height. Unlike $UVPM_{pri}$ profiles, the vertical distributions of secondary UVPM

($UVPM_{sec}$) were more uniform, and the differences among the profiles were more

significant than $UVPM_{pri}$ profiles. Therefore, structure of UVPM profiles were

dominated by the secondary formation. More interestingly, the peak of $UVPM_{sec}$

profiles was getting closer to the ground from 11:00 to 23:00, which may be related to

regional transport of the secondary carbonaceous aerosols and then downward

invasion by strong mechanical turbulence (Zhao et al., 2023), which will be discussed

in details in the following sections.

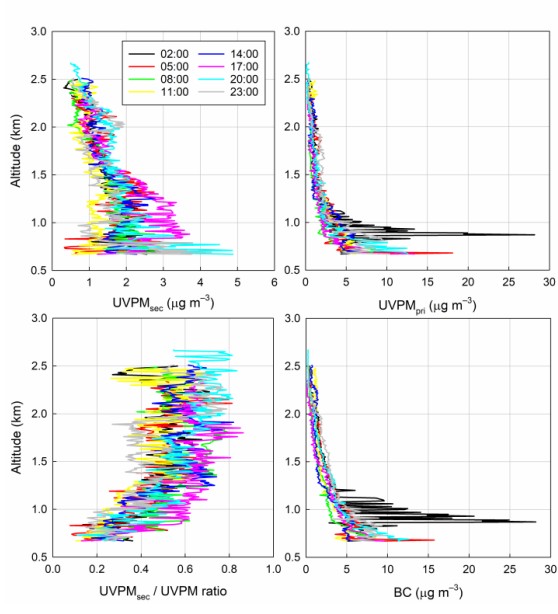



Fig. 2 Diurnal variations of vertical profiles of $UVPM_{sec}$, $UVPM_{pri}$, BC and $UVPM_{sec}$ / UVPM ratio during the campaign.

In order to better understand the mechanisms of the more uniform profiles of $UVPM_{sec}$ as compared to those of BC and $UVPM_{pri}$, we firstly analyzed the relationships between $UVPM_{sec}$ or $UVPM_{pri}$ and BC (Figs. 3 and S1). The $UVPM_{pri}$ concentrations linearly increased with BC at different times of the day (correlation coefficients higher than 0.99). The correlation coefficients in conjunction with

coefficients of divergence (COD) are considered to better characterize the similarity of sources and the uniformity of pollutant concentrations (Wilson et al., 2005). The high correlation between BC and $UVPM_{pri}$ suggested that they shared similar sources, which does not necessarily indicate uniformity. A COD of zero means there are no differences between concentrations of the pollutants, while a value approaching one

indicates maximum differences. A moderate difference is observed during the campaign on the basis of high COD values (0.108–0.179) at 02:00–11:00 and 23:00, indicating there are limited similarities between BC and $UVPM_{pri}$ at the times of the day. Specifically, the differences between BC and $UVPM_{pri}$ are getting smaller and smaller with the increasing altitudes at 02:00–11:00 and 23:00, while those are

independent on altitudes with the low COD values (0.039–0.098) at 14:00–20:00 (Fig. S1). Diurnal variations of the differences between profiles of BC and $UVPM_{pri}$ are closely related to high PBL height and strong turbulent diffusion at noon and afternoon. $UVPM_{sec}$ concentrations nonlinearly varied with BC at different times of the day (Fig. 3). During the daytime, $UVPM_{sec}$ firstly increased with BC

concentrations and then decreased gradually as the increased BC. The synchronous increases of $UVPM_{sec}$ and BC indicated that the low concentrations of primary emissions are favorable for secondary formation, while more primary particles inhibited secondary formation by a series of processes, such as coagulation of new particles by the large particles and scattering solar radiation. The relation of $UVPM_{sec}$

and BC is not significant during the nighttime due to the weaker secondary formation in the absence of the sun.





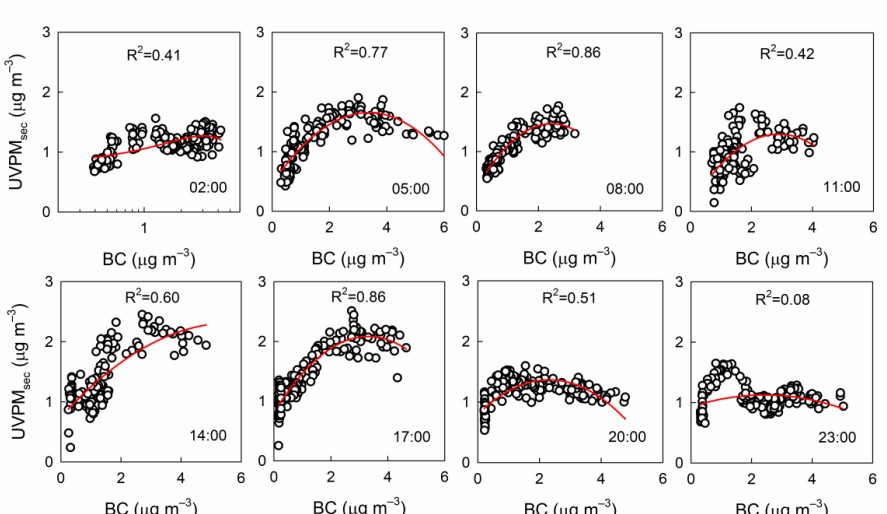

Fig. 3 Diurnal variations of the relationships between the profiles of BC and UVPM$_{pri}$ during the campaign. The relationships were fitted with the binary linear regression (red lines) and coefficients of determination ($R^2$) were given in each subplot.

The correlation between UVPM$_{sec}$/UVPM ratio and UVPM$_{sec}$ is weak with the coefficients of determination ranging from 0.03 to 0.38 at the whole layer (Fig. S2). Therefore, to further investigate the relationships between primary emissions and secondary formation and in consideration of planetary boundary layer (PBL) development, Fig. S3 showed the relationships between UVPM$_{sec}$/UVPM ratio and UVPM$_{pri}$ or UVPM$_{sec}$ concentrations at the varying altitude ranges at the different times of the day. Generally, the ratio increased with the strong secondary formation and decreased as the raised primary emissions. Within PBL, the primary UVPM is much higher than the secondary UVPM (the first column of Fig. S3), while the secondary formation is much stronger at the higher altitudes, resulting in the ratio significantly increased with elevation. Additionally, the relationships between UVPM$_{sec}$/UVPM ratio and UVPM$_{sec}$ are much stronger at the upper air (the coefficients of determination of 0.37 to 0.94) than those within PBL (the coefficients of determination of 0.10 to 0.67), while the correlation between the ratio and UVPM$_{pri}$ is the stronger at the low-level air. The above phenomenon also indicated that less





particles are helpful for the secondary formation of aerosol particles.

### 3.2 Impact of long-range and regional transports on aerosol vertical profiles

The long-range transport can largely impact on the vertical distributions of air pollutants, especially inside the basin terrain (Huang et al., 2008; Zhang et al., 2022). The pollutants originating from the surrounding mountain can be transported to the upper basin by multi-scale circulation, such as mountain-plain winds and valley winds, the transported pollutants impacted basin environment by aerosol-PBL

feedbacks (Zhao et al., 2023). To identify the difference of impact of long-range transport on both BC and UVPM, we compared gridded back trajectory concentrations showing mean UVPM and BC concentrations using the CWT approach at 100 m, 700 m and 1300 m AGL during the campaign (Fig. 4). The gridded back trajectory concentration indicated that the high UVPM and BC

concentrations at the three heights potentially originate from South Asia and central SCB. There is no significant difference for the potential source regions of BC and UVPM at each height. Therefore, the difference of vertical structures of both primary and secondary pollutants in Fig. 2 may be independent on long-range transport.

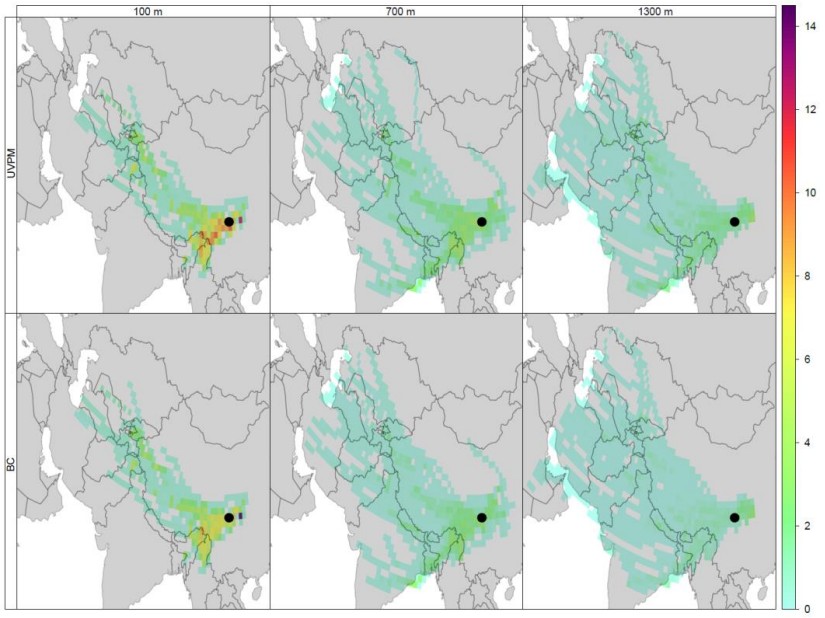



Fig. 4 Gridded back trajectory concentrations showing mean UVPM and BC
concentrations using the CWT approach at 100 m, 700 m and 1300 m AGL during the
campaign.

BC and UVPM pollution rose at the three heights also were checked to see the impact
of regional transport (Fig. 5). At 100 m AGL with the highest BC and UVPM mass
concentrations, the highest BC and UVPM corresponds to southwesterly and
southeasterly winds. At 700 m AGL, the locations of severe BC and UVPM pollution
toward the experiment site vary from southwest and south to northeast, while at the
higher altitude of 1300 m AGL, the BC and UVPM origination relative to the site is
mainly at northeast. Briefly, the UVPM pollution rose is consistent with BC at the
three heights, and thus the regional transport is also not a key factor modulating the
different vertical profiles of BC and UVPM mass concentrations during the field
campaign (see Fig. 2).

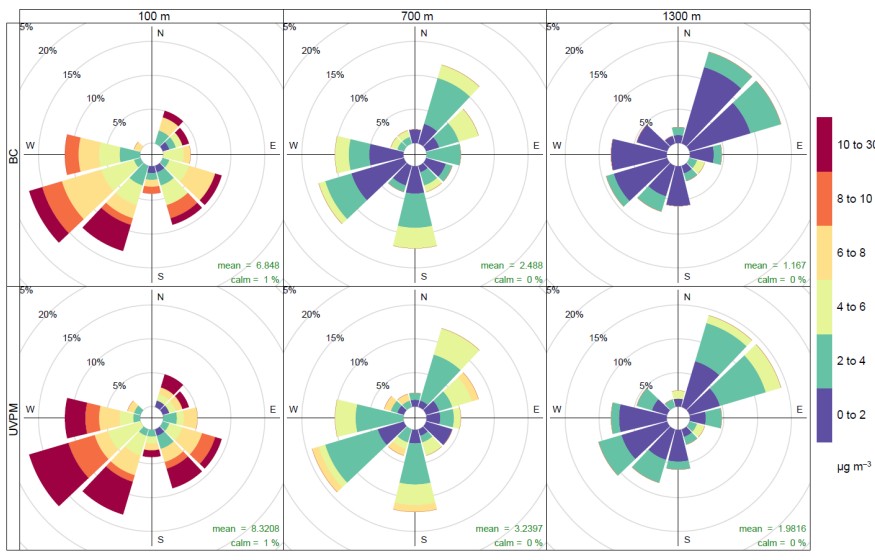

Fig. 5 BC and UVPM pollution rose at 100 m, 700 m and 1300 m AGL during the
campaign. Mean BC and UVPM mass concentrations also were given in the
corresponding subplot.

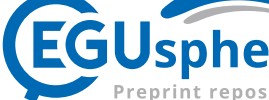

### 3.3 Impact of downward invasion of upper air

To extract more valuable information, the obtained $UVPM_{sec}$ profiles were divided

into three groups during the campaign. The corresponding mean vertical profiles of

other pollutants (BC, $SO_2$, $O_3$ and $PM_1$) and the meteorological factors (T, $w$, $S_{vec}$, $S_{dir}$,

and $V_{TKE}$) also were calculated and showed in Fig. 6, and the frequency of each

cluster at each observation hour of the day and sequence of the three clusters were

showed in Fig. 7. The profile structure and magnitude of $UVPM_{sec}$ exist significant

difference among the three clusters. Cluster 1, accounting for 17.28% of all profiles,

mass concentrations of $UVPM_{sec}$, BC and $PM_1$ weakly fluctuate at the vertical

direction with a weak peak of $UVPM_{sec}$ at ~ 2.0 km. Compared with Clusters 2 and 3,

atmospheric stratification is more unstable in response to the much larger difference in

temperature between the low-level and upper air (Fig. 6f), leading to the stronger

ascending motion below 2.0 km above sea level (ASL). The vertical wind shear ($S_{vec}$,

$S_{dir}$) and mechanical turbulence ($V_{TKE}$) are weaker as compared with that for Clusters

2 and 3, which may be mainly related to occurrence in the nighttime for Cluster 1

(Fig. 7a). $O_3$ concentrations were low due to weak photochemical reactions during

nighttime. Therefore, the more uniform $UVPM_{sec}$ profiles during nighttime are mainly

modulated by thermodynamic processes (temperature).

Cluster 2, comparable frequency of Cluster 1 (16.05%), the primary PM pollution

(BC, $PM_1$) is the lightest, while $UVPM_{sec}$ below 1.7 km ASL (Fig. 6a) and $O_3$

throughout the whole layer (Fig. 6d) are the severest among the clusters. Compared

with Cluster 1, $UVPM_{sec}$ concentration below 1.7 km ASL is much higher with an

obvious peak around 1.4 km ASL, while above the height, it rapidly reduces to below

0.3 µg m$^{-3}$ at ~ 2.5 km ASL, which is the lowest among the clusters. From the

meteorological factor perspectives, the temperature is much lower than Cluster 1 and

is comparable to Cluster 3 at the whole layer. Unlike Cluster 1, the subsiding motion

is throughout the whole layer with the strongest at 2.0–2.5 ASL, and wind shear ($S_{vec}$,

$S_{dir}$) and mechanical turbulence ($V_{TKE}$) are significantly stronger at upper air, which

may be closely related to its appearance in the daytime (Fig. 7a). Therefore,



combining the vertical profiles of primary and secondary pollutants with

meteorological factors, it is inferred that the rapid reduction in UVPM$_{sec}$ with the

increasing altitude for Cluster 2 is mainly controlled by dynamic processes (wind and

turbulence).

Cluster 3, accounting for two-thirds of the profiles, is the most frequent during the

campaign (66.67%). The cluster appears uniformly throughout the day ranging 9% to

15%. For the cluster, secondary UVPM concentrations below 2.0 km ASL and SO$_2$

throughout the whole layer are the lowest among the three clusters. Similar with

Cluster 2, there is weak ascending motion below 1.0 km ASL and gradually converts

to subsiding motion with the increasing altitude to reach the maximum intensity at 2.0

km ASL, and the vertical structure and magnitude of mechanical turbulence index also

is comparable to Cluster 2. The dynamic processes (descending motion and

mechanical turbulence) are comparable between Clusters 2 and 3, but the vertical

profile of UVPM$_{sec}$ is more uniform for Cluster 3 due to the relatively lower UVPM$_{sec}$

concentration at the upper air, i.e., lack of material sources. Therefore, Cluster 3

represents the background profile of UVPM$_{sec}$ at the observation site during the

campaign.

As the previous mentioned, long-range and regional transports also modulate the

vertical structure of air pollutants. Therefore, we checked UVPM$_{sec}$ pollution rose and

gridded back trajectory concentrations showing mean UVPM and BC concentrations

using the CWT approach for the three clusters during the campaign (Figs. S4 and S5).

As shown in the two figures, the regional transport exists some difference among the

clusters, but the highest UVPM$_{sec}$ concentrations mainly correspond to the northerly

winds. For each cluster, there is small difference between UVPM and BC for the

gridded back trajectory concentrations, and UVPM and BC mainly originated from

South Asia and SCB. Therefore, the discrepancy among the three clusters is less

influenced by regional and long-range transports, while it is more modulated by PBL

meteorological processes, such as thermodynamic and dynamic processes.





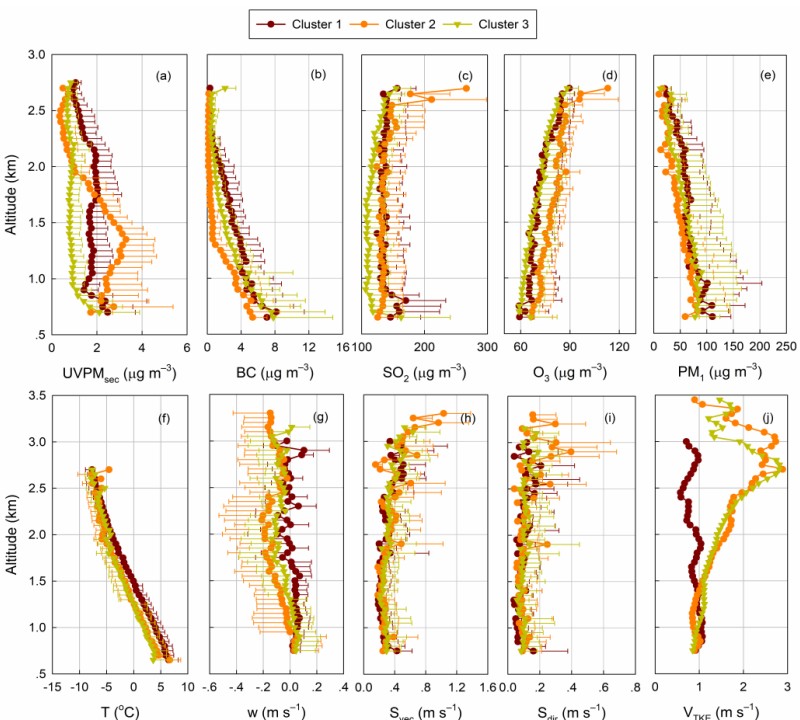

Fig. 6 (a) Three clusters of typical $UVPM_{sec}$ profiles and (b–e) the corresponding mean vertical profiles of the other air pollutants (BC, $SO_2$, $O_3$ and $PM_1$) and (f–j) the mean vertical profiles of temperature (T), vertical velocity ($w$), vector shear ($S_{vec}$), wind-directional shear ($S_{dir}$) and mechanical turbulence index ($V_{TKE}$) during the field campaign. The error bars showed the standard deviation among the profiles within groups at the specific height.

Based on the above analyses, we also checked the occurrence sequence of Clusters 1–3 (Figs. 7a–7d), such as the occurrence frequency of Clusters 2 and 3 at 3–21 hours with three-hours intervals after occurrence of Cluster 1 etc., which was used in the study of Zhao et al. (2021). As shown in Fig. 7a, after the occurrence of Cluster 1 (mainly appear in the nighttime), the frequency of Cluster 2 (mainly occur in the daytime) gradually increases within 3–21 hours. Therefore, it is inferred that the nighttime $UVPM_{sec}$ above 1.7 km ASL downward invades to PBL to induce the significant $UVPM_{sec}$ peak in the daytime with the development of mechanical



turbulence and wind shear. Zhao et al. (2023) also found that mechanical turbulence largely modulates the vertical profiles of air pollutants, which can confirm our findings. After Cluster 2, Cluster 1 is more frequent (Fig. 7c), and thus daytime $UVPM_{sec}$ within the PBL is gradually dispersed to the upper air by thermodynamic processes. Clusters 1 and 2 appear alternately during the campaign. After Cluster 3, the frequency of Clusters 1 and 2 is comparable, and the occurrence of which cluster is dependent on daytime or nighttime.

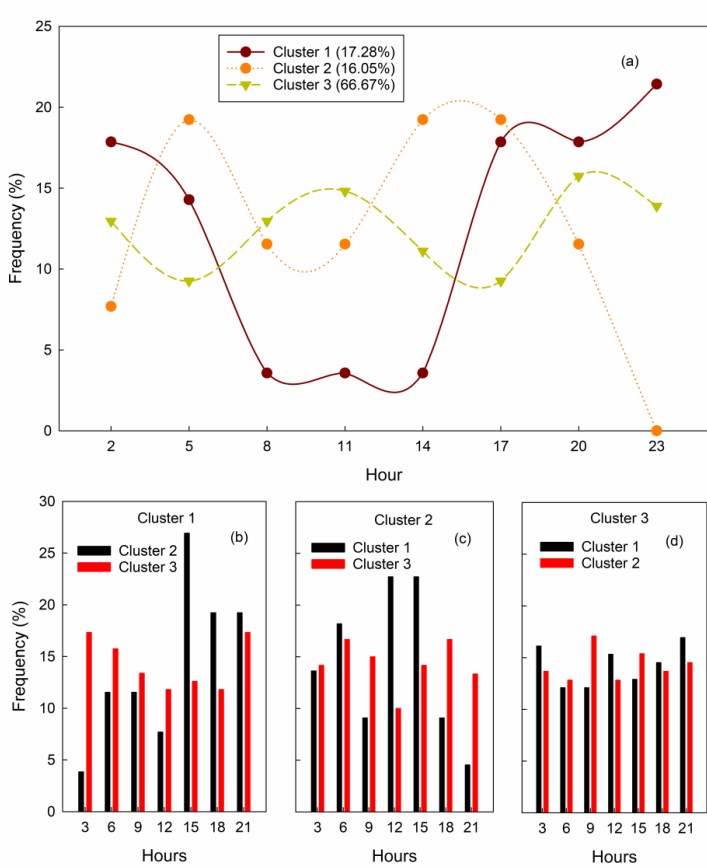

Fig. 7 (a) Diurnal variations of occurrence frequence of the three clusters of vertical profile and (b–d) occurrence frequency of the other two clusters at 3–21 hours with three-hours intervals after occurrence of the specific cluster.

We selected two typical cases on 1 and 7 January 2019 (Figs. 8 and S6) to better



explain the impact of thermodynamic and dynamic processes on the vertical distribution of air pollutants. On 7 January 2019, compared with rapid reduction in

BC and $PM_1$ concentrations with the increasing altitude, $UVPM_{sec}$ concentrations change much weaker at the vertical direction with two obvious peaks near the ground surface and at 2.0–2.5 km ASL, and the location of peak varies with the time of day. From the thermodynamic processes (temperature) perspectives, the surface temperature inversion has a certain impact on the peak of air pollutants near the

ground surface. However, the temperature is comparable at the upper air among the hours of the day, and thus which factors modulate the upper peak should be deeply studied. Unlike temperature profiles, the profiles of vertical velocity and wind shear (Sh, $S_{vec}$ and $S_{dir}$) exist large difference among the times of the day. Furthermore, the $UVPM_{sec}$ peaks well correspond to the strong descending motion and wind shear, and

thus the $UVPM_{sec}$ peaks at the upper air on 7 January 2019 are mainly modulated by dynamic processes instead of thermodynamic processes.

Similar with the previous case, the much rapider reduction of BC than $UVPM_{sec}$ with the increasing altitude on 1 January 2019. However, structure of $PM_1$ profiles is

dominated by $UVPM_{sec}$ rather than BC. In the early morning and late evening, air pollutants are trapped below 1.0 km ASL due to the impact of temperature inversion near the ground surface induced by radiative cooling (Figs. S6a–e). As the surface is heated up and PBL developed during the daytime, the peaks of $UVPM_{sec}$ and $PM_1$ mass concentrations gradually get farther and farther away from the ground, and $PM_1$

peak reaches 2.3 km ASL at 17:00. Thereafter, location of the peak rapidly lowers and the magnitude increases significantly at 20:00. From the perspectives of vertical velocity and wind shears, the subsiding motion is significant above 1.5 km ASL throughout the day, while the ascending motion is obvious below 1.5 km ASL at 17:00. Furthermore, the wind shears (Sh, $S_{vec}$, and $S_{dir}$) are stronger above 1.5 km ASL

at the daytime (11:00–17:00). Therefore, the variations of $UVPM_{sec}$ and $PM_1$ peaks at 1.5–2.0 km above ASL are jointly by both thermodynamic processes from the ground surface and dynamic processes from the upper air (Zhao et al., 2023), while the peaks



near the ground surface are mainly modulated by thermodynamic processes. Zhao et al. (2023) also found that vertical structure of UVPM and BC is largely different

within SiChuan Basin, and thermodynamic and dynamic processes were used to explain the phenomenon, which can better support our findings.

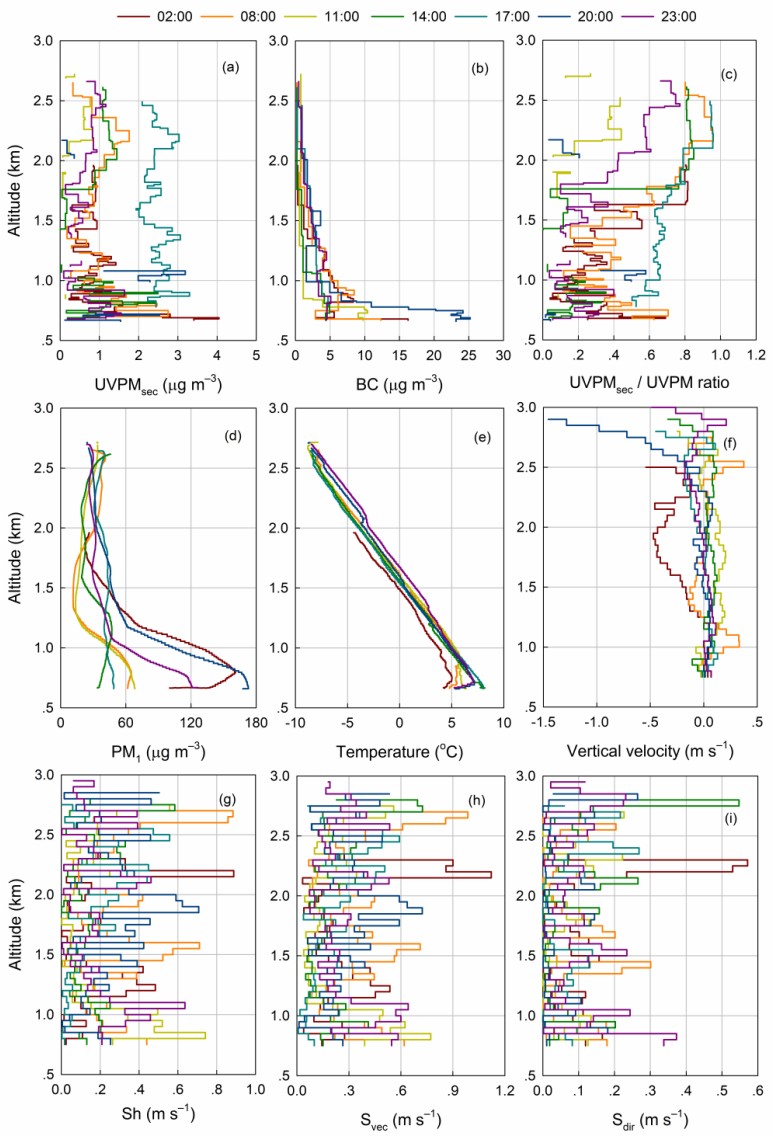

Fig. 8 Diurnal variations of air pollutants (UVPM$_{sec}$, BC, UVPM$_{sec}$/UVPM ratio, and PM$_1$) and meteorological factors (temperature, vertical velocity, Sh, $S_{vec}$, and $S_{dir}$) on 7

January 2019.



## 4   Conclusions

The first Boundary-Layer Meteorology and Pollution at SiChuan Basin (BLMP-SCB) was conducted from December 2018 to January 2019 to deeply understand the interactions between meteorology and pollution at the complex terrain. The vertical

profiles of temperature, RH and air pollutants (CO, NO, $NO_2$, $O_3$, TVOC, BC, UVPM) were observed every three-hours by the instruments carried by tethered balloon. A Doppler Wind Lidar (Windcube 200s, Leosphere, France) was used to obtain the profiles of winds (wind speed and direction, vertical velocity). Based on the data from BLMP-SCB, this study analyzed the impact of mechanical turbulence and

wind shear on vertical profiles of air pollutants. Some novel findings were obtained as follows.

The primary PM (BC, $UVPM_{pri}$) concentrations reduce rapidly with the altitude, while the reduction of secondary PM ($UVPM_{sec}$) concentrations is the slower and even

occurs high values at 1.5–2.0 km ASL, which is explained from the two perspectives of regional transport and downward invasion of upper air in this study. There is small difference for the backward trajectories and pollution rose between BC and UVPM, and thus the discrepancy of vertical structure between BC and UVPM cannot be attributed to regional transport. Combining the clustering analysis technique with case

study, the thermodynamic processes (temperature) are found to be dominant factors for the nighttime uniform UVPM profiles. However, at the daytime, the secondary air pollutants at upper PBL can downward invade into PBL by the strong dynamic processes (mechanical turbulence and wind shear), resulting in highly secondary pollutants at 1.5 km above sea level. The study is significant for deeply understanding

the formation mechanism of unique profile of air pollutants and expanding the interactions between meteorology and pollution at the complex terrain.

We obtained some novel findings, while there are some limitations for this study. The field campaign of BLMP-SCB was conducted at only a rural site of eastern foothills

of Tibetan Plateau, and the observation period was too short to obtain more solid



conclusions. Therefore, we will conduct the second Boundary-Layer Meteorology and Pollution at SiChuan Basin (BLMP-SCB II) at a rural site of Yaan City, more southern than the site of the previous campaign. Combining the data from the two filed campaigns, we hope to get some more universal laws for the interactions between

meteorology and pollution at the complex terrain, especially at the sloped terrain from the basin to Tibetan Plateau. The universal conclusions are important for understanding the formation mechanism of heavy air pollution and then specifying the corresponding countermeasures.

**Code/Data availability**

The code and data used in this work can be accessed by contacting the corresponding author.

**Author contribution**

Suping Zhao and Ye Yu designed the field experiments and Tong Zhang, Guo Zhao and Shaofeng Qi carried them out. Jianjun He instructed the works. Longxiang Dong and Yiting Lv analyzed the relevant data. Suping Zhao prepared the manuscript with contributions from all co-authors.

**Acknowledgements**

This work was supported by the National Natural Science Foundation of China (42422504), Major Science and Technology Project of Gansu Province (24ZD13FA003), and Excellent Member of Youth Innovation Promotion Association, Chinese Academy of Sciences (Y2021111), and Youth United Funding of Lanzhou

Branch of Chinese Academy of Sciences.

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
