# Peer review of "Upper-air secondary pollutants downward invade to planetary boundary layer"

_EGUsphere, 2025_

## Referee Comment (RC1)

Dear Authors,

Below you'll find many comments on this article. I want to make one thing clear, all comments are aimed strictly at the writing and the science and not at any of the authors. With that, if any comment feels like a personal attack I wish to extend my sincere apologies as that was not the intention. My research focus is on turbulence and droplets in the surface layer so some of my comments about aerosols may be due to a lack of knowledge in that specific area but should still be addressed.

This article needs significant work to be accepted. I think the work is interesting so I will recommend major revisions. The revisions that are necessary will make the article look like a completely new submission.

**General Comments**

The English writing in this article is a bit subpar. I understand that English is not everyone's first language but there are many instances where the language used detracts from the science. I have made several comments about the language under the "Minor Issues" section but it is not an exhaustive list. I highly recommend the authors have the article proof-read by a person proficient in English grammar to improve readability.

The introduction need significant rewriting for clarity. Most of the issues are English grammar related, but there are some issues of flow. It is unclear what the hypotheses are and what questions the authors are trying to answer. Black and brown carbon are brought up but the $UVPM_{sec}$ that are mentioned in the abstract are not mentioned.

The Data and Methods section also needs significant rewriting. I have outlined some specific comments below but it is hard to follow. Consider splitting this section up in to something like:

1. Experimental setup and Data Collection
   1.1. Instrument specific quality control
2. Methods
   2.1. Specific methods used get a subsection

There are significant deficiencies in the turbulence section of this article (Section 2.5). Terms are used that I have not seen in the atmospheric literature and no references are given to the way the terms are defined. Since aerosols and turbulence are so interlinked, I strongly encourage the authors to look at how turbulence statistics are used in similar articles.

Due to the poor English writing my comments on the Results section focus mainly on the figures.

Things to consider (in no particular order):
- Better description of experimental setup

- How are tethered balloon profiles taken? Is it up and down at each timestamp (Line 140)? Or is the balloon constantly up with an array of radiosondes along the tether length.
  - If up-down are the profiles shown an average of the up and down sounding?
  - How long was each up and down sounding?
  - Did the conditions change significantly within a sounding?
  - Are the lines of Fig. 2 averages of all 40 days? Or is it a single day?
- Getting to the end of the paper I still do not understand the difference between the variety of UVPM particle types mentioned.
- Supplemental information, at least to me, should be standalone and not referenced significantly in the actual article. Currently the figures in the supplemental information are talked about as if the reader has them on the page in front of them.
- Figures are hard to interpret, especially the supplemental ones.
- Make very clear what data are used and where.
- Define the three groups (clusters???) early on in the article
- Keep figure style consistent throughout to make comparisons easy
- The flow of the article is nonexistent. I do not have a good recommendation at the moment but the authors need to sit down and outline the article first and make sure the discussion and results come in both a logical and aesthetic way. The article is rather lengthy and I feel like another pass at an outline will help the authors remove unnecessary information.
- The results don't really show me how turbulence has any impact on the vertical profile of aerosols.

**Major Issues**

*Figures (in general):*  The authors talk about daytime vs nighttime and impact of altitude on concentrations. However, all of the figures are very hard to interpret since there is no clear separation between day vs night and altitudes. Fig. S3 has some altitude information but it is nigh on impossible to understand. I strongly recommend the authors organize their figures such that differences due to day, night, and altitude are very clear. Along with any other key parts the authors wish to highlight

*Line 146-149: "The performances of the…within PBL."* – this sentence is rather unclear. What sensors were validated? What were the reference instruments? A little bit more information here would be very helpful, yes there is a reference but checking a reference to understand the meaning of a sentence creates very poor readability.

*Figure 1: -* This figure is not very helpful to understand the experimental setup. A table describing the instrumentation might be more helpful. Further it is unclear from Fig. 1 if there are multiple aerosol samplers at different elevations along the Tibetan Plateau. I would recommend a 2-D map showing location of all sites and then a table describing the equipment at each site. Further since the authors are talking about gradients some information on the measurement heights is crucial.

*Line 194-195: "…Hybrid Single-Particle Lagrangian Integrated Trajectory…"* – what is this model? Did you make it or did it already exist? If it exists a reference would be needed. If you created it then more detail is necessary

*Line 203: "…combining BC and UVPM…"* – I'm still not entirely sure what UVPM is as it was poorly described in the preceding sections.

*Section 2.3* – A more thorough description here would be very helpful, perhaps with a figure. You say you used cluster analysis to divide the UVPM profiles in to three groups– this would be a great section in which to introduce those three groups with a table so that the reader knows what to expect in the latter sections.

*Section 2.4:* - This section seems like it needs to be higher up – see my note about organizing this entire section under "General Comments"

*Line 237:* - UVPM Ratio is undefined

*Line 236: "…the fraction of UVPM$_{sec}$ in UVPM."* – UVPM is very poorly defined, I thought all UVPM was 'secondary'? More care needs to be taken to describe these particle types in the introduction and data collection sections.

*Equation 1 & 2:* It is now very unclear to me now what aerosol types you are measuring and which types you are estimating through relationships. Further now there is a UVPM$_{pri}$ designation which is not defined (first shows up on Line 237).

*Equation 3:* Is "mechanical turbulence index" a term the authors came up with? This is the square root of the turbulent kinetic energy (TKE) which is like some root mean square velocity. Further TKE encompasses all turbulence and not just shear which is what the authors are trying to isolate. I strongly recommend the authors look at the TKE budget equation and go from there.

*Equation 4:* This notation is very unclear. Shear is a derivative so should look something like $\frac{du}{dz}$ where $u$ is the streamwise wind speed.

*Equation 5:* I do not understand what this equation gives. How are you defining your streamwise (u) and spanwise (v) velocities? Traditionally $\overline{v} = 0$ since the coordinates are rotated such that the mean wind speed $U = \overline{u}$ and $\overline{v} = \overline{w} = 0$ where $\overline{u}$ is the streamwise wind velocity, $\overline{v}$ is the spanwise wind velocity, and $\overline{w}$ is the vertical wind velocity.

*Figure 2:* Text is very small and colors are random. In an earlier section the authors mentioned a difference in profiles between day and night. It would make sense to separate these figures based on day and night. Having the colors be a gradient from start time to end time would greatly improve readability. The authors could also consider a height time concentration plot. See example below except instead of particle size on the y-axis you will have your elevation above ground. It is also unclear how these profiles were taken, was the tethered balloon raised to 2.5 km above ground level and back down at each time stamp?  There are no labels on the panels of this figure, and the caption does a poor job of

describing what the reader is looking at. I've also included a caption below the example figure so the authors can understand what a caption should include

[Figure]

**Figure X:** Time series of bin-wise droplet concentration from the FM120 and FD70. During the periods of fog (0300 – 1200) we see significantly fewer larger particles from the FD70 and towards the end of the period (1100) we see the evaporation of all larger particles which result in the broadening of the peak of small particles.

*Lines 298-305: "The vertical distribution of…discussed in details in the following sections."* – Nearly everything that was written in these lines is impossible to read from Fig. 2.

*Figure S1 & Figure 3:* S1 seems like it would be more useful in the paper instead of in supplemental information. Further, the way both Figure 3 and S1 are presented makes it hard to see any trend. S1 is easier since BC and UVPM$_{pri}$ basically follow each for all time periods but then in Fig. 3 it is unclear if ever the two aerosol types follow each other. Again it would be helpful to separate these plots by day and night since that is what the paper talks about.

*Figure 4:* It is unclear what I am looking at here. No labels are present. It is also unclear how these calculations were done. I'm sure it was a mix of the model from earlier and whatever "CWT" is. It is not clear at all how the differences between BC and UVPM source areas is calculated.

*Figure 5:* UVPM and BC concentrations are flipped with respect to the previous plot making it hard to make inter-plot comparisons. Labels are very hard to read from a comfortable reading distance. It is not clear where the data for this plot comes from.

*Line 395-396: "…three groups…"* – These groups are not defined at all. Because they are not defined it is very hard to follow the remainder of this section.

*Figure 6:* Because the turbulence terms are defined in a way that I do not understand I do not know how to interpret this plot. Why are error bars all of the sudden included here and not on previous plots? I also don't see significant differences in the three clusters except in UVPM$_{sec}$ concentrations.

*Figure 8:* See point for Figure 2.

*Conclusion:* This section is way too vague. "novel findings" may be appropriate for a conference abstract but not so much for an article. The conclusion does a poor job in tying together the entire paper.

**Minor Issues**

*Abstract:* - I've rewritten the abstract to give the authors some guidelines on what kind of grammar mistakes need correcting. Please note that I have simply rewritten the abstract as it stands and the authors should not just use my example directly. I do not think this is a good abstract for an article, this seems more like a vague abstract you would see for a conference presentation. I strongly recommend the authors incorporate their actual results in to the abstract in a more direct way.

Field experiments that are focused on the interactions between aerosols and the planetary boundary layer (BPL) in complex terrain are few and far between. In this article we use data from the First Planetery Boundary-Layer Meteorology and Pollution at the western SiChuan Basin ($1^{st}$ BLMP-SCP) campaign from Dec. 2018 to Jan. 2019. The focus of the campaign is to provide data on the impact of BPL turbulence on the profiles of air pollutants. We focus on two pollutant types: primary particulate matter ($PM_{pri}$) and secondary ultraviolet PM ($UVPM_{sec}$). Both $PM_{pri}$ and $UVPM_{sec}$ show similar regional and long-range transport throughout the campaign. However, the concentration of $PM_{pri}$ rapidly declines with increasing altitude while the concentration of $UVPM_{sec}$ has a peak at 1.5 – 2 km above ground level. We show that the concentration of $UVPM_{sec}$ is more uniform at night and during the daytime $UVPM_{sec}$ gets mixed down into the PBL through mechanical turbulence. This study emphasizes the importance of turbulence on the vertical profiles of air pollutants in complex terrain. Specifically, the results are helpful for understanding the formation mechanism of heavy air pollution within these complex topographic environments.

*Line 32:* "…filed campaign…" – should be "field." There are also several other places where "field" is misspelled and general spelling and grammar issues.

*Line 61:* *"Aerosol-planetary boundary layer (PBL)…"* – While I understand what the authors are saying here it can be a bit confusing. For clarity, I recommend the authors write something along the lines of:

"The interactions between aerosols and the planetary boundary layer (PBL)…"

*Line 68: "BC can also…"* – typically starting a sentence with an acronym is to be avoided for readability purposes. This comment applies to many places in the article.

*Line 79:* "…within PBL are…" – "…within the PBL are…" – this comment applies to all cases where PBL is used, "the" is necessary for readability purposes. Also applies to other cases like "Tibetan Plateau".

*Line 166-174: "A portable GPS…given by the manufacturer"* – Very hard to follow section of text. It just all over the place.

*Line 184: "0.2 and 0.05 Hz"* – earlier in the paragraph temporal resolutions were given in seconds and not hertz. For improved readability it would be good to stick to one or the other.

---

## Referee Comment (RC2)

**Message to the authors**

Please see my attached comments below. This research is interesting, and I do believe that this is meaningful science to the research community. However, given the current state of the manuscript, I believe that major revisions are necessary to be suitable for publication. I recommend that the authors carefully address the comments and suggestions, and then resubmit the manuscript as a new submission.

**General Comments**

Most of my comments pertain to the language presented in the manuscript. I understand that English is not everyone's first language, and I apologize in advance if my remarks come across as insensitive in any way. I strongly recommend that the authors have this manuscript proofread by someone proficient in English. Doing so will help elucidate some of the language used and improve the overall quality of the manuscript. Please note that some aspects of the research presented here are outside my area of expertise, so I apologize in advance if any of my comments seem basic.

Data and Methods: I believe this section needs substantial revision, particularly with regard to:

- Site and instrumentation description
- Data utilization and quality control, in particular:
    - Validation of deployed instrument measurements to ground based instruments
    - Quality control techniques
- Analysis techniques in sections detailing:
    - K-means cluster analysis
    - Utilization of turbulence parameters

Results:

- In the manuscript, some figures are referenced with variables that do not appear in the figures themselves. For example, the manuscript mentions a height reference to Figure S1, but the figure does not contain any height dimension.

- I believe that some figures in the supplement should be included in the manuscript itself. For example, Figure S1 is referenced frequently in alongside Figure 3. It makes it easier to read and understand when both figures are present the manuscript rather than having to reference the supplement section for figures.

While these points are discussed in greater detail in the sections below, they highlight a few overarching points that I think should be addressed to the authors of the manuscript.

**Major comments**

Abstract (Lines 32 - 50): Abstract needs to be rewritten, the language makes it a bit hard to follow. There are grammatical and sentence structure issues that I believe after resolving, can really improve the flow of the abstract and elucidate the content for the reader.

All Figures in manuscript: Text on the figures are too small and hard to look at initially for information. For example: Figures 4, 5, S4, and S5 have embedded text that is critical for interpreting the figures, but the small text make this difficult. I recommend enlarging the text in all figures for better clarity.

Lines 134 - 135: "*The first field campaign of Boundary Layer Meteorology and Pollution at SiChuan Basin (BLMP-SCB) was conducted at a rural site (Sanbacun, 103°40'38" E, 30°54'59" N) of eastern foothills of Tibetan Plateau in winter of 2018, lasting about 40 days (Fig. 1)."* - Please include elevation for those not familiar with the region.

Lines 146 - 147: "*The performances of the sensors were verified by comparing with on-ground reference instruments (Pang et al., 2021) ...*" - What instruments specifically? Can you provide some statistics describing these differences? (For example: $R^2$)?

Lines 168 - 170: "*The radiosonde has been widely used and validated (Haman et al., 2012), and there is a very slight difference with the other radiosondes such as 170 Vaisala RS92 (Trapp et al., 2016)."* – Validated how? Again, I think providing quantitative statistics that describe the difference would be beneficial.

Lines 217 – 219: "*Clustering analysis was used to divide the UVPM profiles during the campaign into three groups with comparable vertical structure of*

*UVPM within groups.*" – What are the groups? Explain in greater detail what each group signifies. How did you decide that three clusters were appropriate for this analysis? I suggest providing some sort of clustering validation to determine the correct number of clusters, such as "The Elbow Method" or even "Silhouette score" could be beneficial. I recommend the authors look into this and provide context, you do not necessarily need to provide plots, but I think a simple reference would suffice.

Lines 231 – 233: *"The ONA algorithm results in significant noise reductions and much more reasonable temporal changes in mass concentrations of carbonaceous particles (Cheng and Lin 2013; Park et al., 2010)."* – What exactly is the "ONA algorithm"? I think a more in-depth description into the mechanics and procedures of this algorithm is necessary. What percentage of data required the ONA technique? I think including this is also necessary to include in the manuscript for context.

Line 237: The authors mention $UVPM_{pri}$, but do not reference what it is or what it means? I assume this means organic carbon from primary sources, but this is not directly stated in the prior sections.

Lines 252 – 254: At this stage in the manuscript, the distinction between UVPMpri and UVPMsec in Equations 1 and 2 is not clear to me. In Eq. 2 UVPM is referenced, but the manuscript does not specify how UVPM differs from UVPMpri and UVPMsec.

*Section 2.5 Calculation of mechanical turbulence and wind shear*: It seems like the authors want to isolate shear as a main contributor to atmospheric exchange and transport. However, it is not clear to me how this is directly achieved. The authors mention the TKE equation and shear terms, but I recommend the authors look into the TKE Budget equation, if possible, given the instrumentation deployed and measurements available. The TKE budget equation provides more detailed insight into the generation, transport, and dissipation of turbulence across a specific transect.

Lines 310 – 312: *"In order to better understand the mechanisms of the more uniform profiles of UVPMsec as compared to those of BC and UVPMpri, we firstly analyzed the relationships between UVPMsec or UVPMpri and BC (Figs. 3 and S1)."* - This requires a bit of rephrasing to understand. The authors reference Figures 3 and S1 frequently in this section. I suggest moving Figure S1 into the

main text. This just makes it easier to reference especially in a section of the results where both figures are needed together to clarify the interpretation.

Lines 323 – 326: *"Specifically, the differences between BC and UVPMpri are getting smaller and smaller with the increasing altitudes at 02:00–11:00 and 23:00, while those are independent on altitudes with the low COD values (0.039–0.098) at 14:00–20:00 (Fig. S1)."* – Authors mention altitude, but altitude is not on Figure S1.

Lines 329 – 330: *"During the daytime, UVPMsec firstly increased with BC concentrations and then decreased gradually as the increased BC."* – Rephrase, this is a bit confusing for the reader to understand.

Lines 493 – 496: *". Furthermore, the UVPMsec peaks well correspond to the strong descending motion and wind shear, and thus the UVPMsec peaks at the upper air on 7 January 2019 are mainly modulated by dynamic processes instead of thermodynamic processes."* – Not sure what "peak" is referenced here, not clear by looking at the figure(s).

Lines 502 – 503: *"As the surface is heated up and PBL developed during the daytime…"* – I would suggest adding a plot that shows a time series of PBL development with height either from the Lidar or other remote sensing instruments (if possible). This helps visualize how deep and fast the PBL grows throughout the day.

**Minor Comments:**

All figures: Figures that reference time (For example: Fig 3), are these all in local time? UTC?

Line 130: *"… understanding the change in air pollutants and then taking targeted measures."* – What do you mean by "taking targeted measures"?

Lines 173 – 174: *"The uncertainty of temperature and RH measurements was ±0.3 ℃ and ±5% given by the manufacturer."* – Provide citation of manufacturer

Line 183: *"The DBS mode was used in this campaign."* – What is "DBS mode"? I suggest the authors describe this in greater detail. Provide citations.

Lines 189 – 191: What instruments were on the tower? Can you provide model #'s, also basic statistics on comparison between Lidar and sonic wind anemometer values?

Lines 298 – 300: *"Unlike UVPMpri profiles, the vertical distributions of secondary UVPM (UVPMsec) were more uniform, and the differences among the profiles were more significant than UVPMpri profiles."* – This is a bit unclear and should be rephrased for clarity.

Lines 346 – 348: *"Fig. S3 showed the relationships between UVPMsec/UVPM ratio and UVPMpri or UVPMsec concentrations at the varying altitude ranges at the different times of the day."* – Rephrase for clarity. What do you mean by "UVPMpri or UVPMsec"? Do you mean: "Figure S3 showed the relationships between the UVPMsec/UVPM ratio and UVPMpri, as well as between the UVPMsec/UVPM ratio and UVPMsec."?

---

## Author Comment (AC1)

**Responses to the Comments of Reviewer #1**

We express our sincere gratitude for your invaluable suggestions, which are highly beneficial for enhancing the quality of our manuscript. We have meticulously addressed all the points you raised and provided a point - by - point response to each comment as follows.

**General Comments**

1. The English writing in this article is a bit subpar. I understand that English is not everyone's first language but there are many instances where the language used detracts from the science. I have made several comments about the language under the "Minor Issues" section but it is not an exhaustive list. I highly recommend the authors have the article proof-read by a person proficient in English grammar to improve readability.

**Response:** Thank you for your constructive suggestions for improving the manuscript. In accordance with your specific remarks regarding the language within the "Minor Issues" section, the article was meticulously proofread by an individual well - versed in English grammar to enhance its readability. Notable enhancements are discernible across the entirety of the revised manuscript.

2. The introduction need significant rewriting for clarity. Most of the issues are English grammar related, but there are some issues of flow. It is unclear what the hypotheses are and what questions the authors are trying to answer. Black and brown carbon are brought up but the UVPMsec that are mentioned in the abstract are not mentioned.

**Response:** I express my sincere gratitude for your constructive proposals regarding the improvement of the manuscript. In accordance with your suggestions, the introduction has been substantially revised to enhance clarity, encompassing issues related to English grammar and certain flow - related problems.

  The vertical distribution of the concentration and optical properties of aerosol particles serves as crucial information for comprehending the interactions between

aerosols and the planetary boundary layer (PBL), as well as the influence of these interactions on the regional climate and environment (Hallar et al., 2021). As the primary light - absorbing aerosols, the vertical distributions of black carbon and brown carbon are crucial for the structure of the PBL and its associated impacts. However, when compared to surface observations, the collaborative vertical observations of meteorological and environmental variables are challenging and limited, especially in complex terrains. The vertical structure of black carbon and brown carbon, along with its formation mechanism, was primarily elucidated by regional transport and vertical mixing. The mechanism was more intricate in the sloped terrain around the Tibetan Plateau (TP). Under the backdrop of rapid warming, black and brown carbon in South Asia and the Sichuan Basin can be transported to the TP due to the strengthened "heat pump" effect of the TP. Previous research has predominantly concentrated on the transport of pollutants from the surface to the upper atmosphere. In contrast, the mechanism of downward transport of pollutants remains inadequately comprehended. Notably, the role of mechanical turbulence induced by wind shear, which is prominent on the leeward slope of large - scale terrain, has received limited attention. Based on the aforementioned process, the introduction will be elaborated upon in the revised manuscript. Moreover, in fact, UVPM is brown carbon; therefore, $UVPM_{sec}$ will be substituted with $BrC_{sec}$ and comprehensively presented throughout the revised manuscript.

3. The Data and Methods section also needs significant rewriting. I have outlined some specific comments below but it is hard to follow. Consider splitting this section up in to something like:

   1. Experimental setup and Data Collection

   1.1. Instrument specific quality control

   2. Methods

   2.1. Specific methods used get a subsection

**Response:** I express my sincere gratitude for your highly valuable suggestions. As you have pointed out, the "Data and Methods" section lacks clarity and coherence. In the revised manuscript, this section will undergo substantial rewriting and

reorganization, and it will be divided into the following sub - sections.

2.1 Experimental setup and data collection

2.1.1 Experimental setup

2.1.2 Data collection and quality control

2.2 Methods

2.2.1 Identification of potential source regions for BC and BrC$_{sec}$

2.2.2 K-means cluster analysis

2.2.3 Utilization of turbulence parameters

4. There are significant deficiencies in the turbulence section of this article (Section 2.5). Terms are used that I have not seen in the atmospheric literature and no references are given to the way the terms are defined. Since aerosols and turbulence are so interlinked, I strongly encourage the authors to look at how turbulence statistics are used in similar articles.

**Response:** Thank you for your constructive suggestions. The turbulence section of this article has undergone substantial revision, and it has been renamed as "2.2.3 Utilization of turbulence parameters", serving as a subsection of "2.2 Methods". The details of the section's revision are presented in blue text as follows.

**2.2.3 Utilization of turbulence parameters**

Turbulent kinetic energy (TKE) serves as an indicator of turbulence intensity, and it represents a crucial variable for the comprehension of the exchanges of energy, water vapor, and greenhouse gases between the land surface and the atmosphere. The turbulent velocity scale, defined as $V_{TKE}=\left[(1/2)\left(\overline{u'^2}+\overline{v'^2}+\overline{w'^2}\right)\right]^{1/2}=\sqrt{TKE}$, has been widely employed to characterize turbulence intensity (Lan et al., 2018; Sun et al., 2012). Here, $u$, $v$, and $w$ denote the zonal, meridional, and vertical wind components respectively, while $u'$, $v'$, and $w'$ signify the standard deviation of each variable. The vertical profiles of $V_{TKE}$ can be derived from the aforementioned equation. A positive correlation exists between the $V_{TKE}$ and mechanical turbulence, such that an increase in $V_{TKE}$ corresponds to an enhancement of mechanical turbulence. The combination of

the profiles of $V_{TKE}$ and air pollutants can be employed to gain a more comprehensive understanding of the downward transport of air pollutants at the eastern foothills of the Tibetan Plateau.

The TKE budget equation offers a more comprehensive and in - depth understanding of the generation, transport, and dissipation of turbulence along a specific transect (Martilli et al., 2002, Eq. 4).

$$\frac{1}{2}\frac{\partial e^2}{\partial t} = -\overline{u'w'}\frac{\partial \overline{u}}{\partial z} + \frac{g}{T}\left(\overline{w'\theta'}\right) - \frac{1}{2\rho}\frac{\partial \overline{(w'e^2)}}{\partial z} - \frac{1}{\rho}\frac{\partial \overline{w'p'}}{\partial z} - \nu \left(\overline{\frac{\partial u_i'}{\partial x_j}\frac{\partial u_i'}{\partial x_j}}\right),\tag{4}$$

where $e$ represents the TKE, $t$ denotes time, $g$ signifies the acceleration due to gravity, $T$ stands for temperature, $\theta$ represents the potential temperature, $\overline{w'\theta'}$ refers to the turbulent heat flux, $\overline{u'w'}$ indicates the turbulent momentum, $\rho$ denotes the air density, $u$ represents the wind speed, $z$ represents the observational height, $w'$ represents the vertical wind velocity fluctuation, and $p'$ represents the atmospheric pressure pulsation. The value of TKE is primarily influenced by mechanical shear and buoyancy effects (represented by the first two terms on the right - hand side of Equation 4). In the context of unstable stratification, a significant correlation exists between TKE and the effects of mechanical shear and buoyancy (Yue et al., 2015). During the campaign, the buoyancy effect is notably suppressed on cloudy days (Song and Zhang, 2024). Furthermore, considering the constraints of the deployed instrumentation and the available measurements with relatively low temporal resolution, this study only analyzed the mechanical shear (including wind speed shear and directional shear).

According to Mahrt (2017), the wind speed shear was defined as

$$Sh \equiv \delta V = \left|V_{(50+i)\,m}\right| - \left|V_{i\,m}\right|,\tag{5}$$

where $\delta V$ denotes the disparities in wind speed ($V$) between adjacent measurement levels with an interval of 50 meters (Fig. S1), and $i$ represents the height at the specific level. Furthermore, $S_{vec}$ was defined as the magnitude of the vector shear,

which was determined by the vertical disparities in the wind - speed components.

$$S_{vec} \equiv \delta V = \sqrt{(\delta u)^2 + (\delta v)^2} = \sqrt{\left(|u_{(50+i)\,m}| - |u_{i\,m}|\right)^2 + \left(|v_{(50+i)\,m}| - |v_{i\,m}|\right)^2}, \qquad (6)$$

where $u$ and $v$ denote the zonal and meridional wind components, respectively. The wind - directional shear can be quantitatively characterized as the disparity between the magnitude of the vector shear and the speed shear.

$$S_{dir} \equiv S_{vec} - Sh, \qquad (7)$$

where $S_{dir}$ is denoted in m s$^{-1}$.

5.  Things to consider (in no particular order):

• Better description of experimental setup

**Response:** Thank you for catching that. The section of "2.1.1 Experimental setup" was better described and the revised Fig. 1 shows experimental configuration for two - dimensional pollution and meteorology within the PBL at the rural site of Sanbacun, which is situated at the eastern foothills of the TP. Table 1 listing equipment utilized and the corresponding parameters during the 1$^{st}$ BLMP - SCB campaign conducted at the rural site was also added to the revised manuscript. The section of "2.1.1 Experimental setup" was showed in blue text as follows.

**2.1.1 Experimental setup**

The First Campaign on Planetary Boundary - Layer Meteorology and Pollution in the Western Sichuan Basin (1$^{st}$ BLMP - SCB) was carried out at a rural location (Sanbacun, 103°40′38″ E, 30°54′59″ N, 650 m) on the eastern foothills of the Tibetan Plateau from December 2018 to January 2019 (during the winter of 2018–2019), with a duration of approximately 40 days (Fig. 1). A tethered balloon, filled with 10 m$^3$ of helium and serving as a carrier with a maximum payload capacity of 8.0 kg, was employed to conduct in - situ observations of the vertical profiles of key PM$_1$ (mass and carbonaceous components), gaseous pollutants (CO, NO, NO$_2$, O$_3$, TVOCs), and meteorological variables (temperature, RH) within the PBL. The balloon ascends or descends at a constant velocity of 0.5 m s$^{-1}$. It is regulated by an electric winch from the ground to reach a specific altitude. When the wind velocity in the upper layer

exceeds 5.0 m s⁻¹, the balloon is retracted to guarantee the safety of the onboard instruments. The variables were monitored at three - hour intervals (02:00, 05:00, 08:00, 11:00, 14:00, 17:00, 20:00, 23:00) as the balloon ascended and descended at each timestamp during the campaign. Each upward and downward sounding endures approximately two hours. The variation in conditions between upward and downward sounding is not substantial. Consequently, the profile at each timestamp represents an average of the upward and downward sounding. Consequently, eight vertical profiles can be acquired daily, which facilitates the comprehension of the diurnal variations of the PBL turbulence and its effects. The local time (Beijing Time = UTC + 8) was employed throughout the manuscript.

[Figure]

Fig. 1 Experimental configuration for two - dimensional pollution and meteorology within the PBL at the rural site of Sanbacun, which is situated at the eastern foothills of the TP. The color transition from green to gray signifies that the altitude spans from 500 m to 4000 m. The equipment employed is presented and detailed in Table 1.

Table 1 Equipment utilized and the corresponding parameters during the 1st BLMP - SCB campaign conducted at the rural site of Sanbacun.

| Instruments | Model | Variables | Accuracy | Time resolution |
|---|---|---|---|---|
| Carbon Analyzer | MA-200 | Carbonaceous aerosols at five wavelengths | $0.001\ \mu g\ m^{-3}$ | 5 seconds |
| Alphasense | $O_X$-B431 | $O_3$ | 1 ppb | 5 seconds |
| Alphasense | $NO_2$-B43F | $NO_2$ | 0.5 ppb | 5 seconds |
| Alphasense | NO-B4 | NO | 5 ppb | 5 seconds |
| Alphasense | CO-B4 | CO | 1 ppb | 5 seconds |
| Alphasense | $SO_2$-B4 | $SO_2$ | 5 ppb | 5 seconds |
| Alphasense | OPC-N2 | $PM_1$, $PM_{2.5}$, $PM_{10}$ | $10\ \mu g\ m^{-3}$ | 5 seconds |
| Alphasense | PID-AH | TVOCs | 1 ppb | 5 seconds |
| Radiosonde | iMet-1-AB | Temperature, RH | $0.3\ ^oC$ for temperature, 5.0% for RH | 1 seconds |
| Wind Doppler Lidar | Windcube-200s | wind speed and direction, vertical wind velocity, and turbulence intensity | $0.1\ m\ s^{-1}$ for speed, and $2^o$ for direction | 5–20 seconds |
| Radiometer | MWP967KV CNGC-01001 | Temperature, RH | 0.8–1.0 K for temperature, 10% for RH | 5 seconds |

• How are tethered balloon profiles taken? Is it up and down at each timestamp (Line 140)?

Or is the balloon constantly up with an array of radiosondes along the tether length.

o  If up-down are the profiles shown an average of the up and down sounding?

o  How long was each up and down sounding?

o  Did the conditions change significantly within a sounding?

o   Are the lines of Fig. 2 averages of all 40 days? Or is it a single day?

**Response:** We appreciate your inquiries. As you have pointed out, there are numerous ambiguous aspects regarding the operations of the tethered balloon in the manuscript. Therefore, these issues have been rectified and more comprehensively elucidated in the revised edition of our manuscript. The relevant contents are presented in the subsequent blue - colored text.

A tethered balloon, filled with 10 $m^3$ of helium and serving as a carrier with a maximum payload capacity of 8.0 kg, was employed to conduct in - situ observations of the vertical profiles of key $PM_1$ (mass and carbonaceous components), gaseous pollutants (CO, NO, $NO_2$, $O_3$, TVOCs), and meteorological variables (temperature, RH) within the PBL. The balloon ascends or descends at a constant velocity of 0.5 m $s^{-1}$. It is regulated by an electric winch from the ground to reach a specific altitude. When the wind velocity in the upper layer exceeds 5.0 m $s^{-1}$, the balloon is retracted to guarantee the safety of the onboard instruments. The variables were monitored at three - hour intervals (02:00, 05:00, 08:00, 11:00, 14:00, 17:00, 20:00, 23:00) as the balloon ascended and descended at each timestamp during the campaign. Each upward and downward sounding endures approximately two hours. The variation in conditions between upward and downward sounding is not substantial. Consequently, the profile at each timestamp represents an average of the upward and downward sounding. Eight vertical profiles can be acquired daily, which facilitates the comprehension of the diurnal variations of the PBL turbulence and its effects. Additionally, the vertical profiles of Fig. 2 are averages of all 40 days.

[Figure]

Fig. 2 Diurnal variations of the vertical profiles of (a) $BrC_{sec}$, (b) $BrC_{pri}$, (c) $BrC_{sec}$/BrC ratio, and (d) BC during the campaign. These profiles represent the averages of all days throughout the campaign. The black arrows in (a) and (c) indicate the temporal movement of the peaks of profiles of $BrC_{sec}$ and the $BrC_{sec}$/BrC ratio.

• Getting to the end of the paper I still do not understand the difference between the variety of UVPM particle types mentioned.

**Response:** Thank you for your question. The expression of UVPM is relatively infrequent; consequently, it was substituted with brown carbon (BrC) in the revised manuscript. The optical and thermal analyses, along with electron microscopy, conducted through laboratory and field experiments, have offered substantial evidence for the presence of certain organic carbon (OC) possessing light - absorbing properties (Kirchstetter et al., 2004). This fraction of absorbing OC, referred to as brown carbon (BrC), exhibits strong absorption in the ultraviolet wavelengths and relatively weaker absorption as it extends into the visible range (Hoffer et al., 2006). Brown carbon is generated through the inefficient combustion of hydrocarbons and the photo - oxidation of biogenic particles (Andreae and Gelencser, 2006), namely primary

sources (BrC$_{pri}$) and secondary formation (BrC$_{sec}$). The above explanations were integrated into the introduction section in the revised manuscript.

• Supplemental information, at least to me, should be standalone and not referenced significantly in the actual article. Currently the figures in the supplemental information are talked about as if the reader has them on the page in front of them.

**Response:** Thank you for your constructive suggestions for improving the readability of the manuscript. As you have indicated, Figures S1, S3 and S6 appears to be more valuable within the paper rather than in the supplementary information. Consequently, The figures have been relocated to the revised main text. In total, six figures (Figs. S1–S6) are incorporated in the revised supplementary information. These figures are standalone and not extensively referenced in the actual article.

• Figures are hard to interpret, especially the supplemental ones.

**Response:** I sincerely appreciate your constructive suggestions. As you have pointed out, the figures presented in the main text and supplementary information are difficult to interpret. Therefore, for the purpose of facilitating better interpretation, all figures have been redrawn and reorganized. Consequently, in the revised version of our manuscript, the differences attributed to daytime, nighttime, and altitude are highly discernible.

• Make very clear what data are used and where.

**Response:** Thank you for your reminder. The First Campaign on Planetary Boundary - Layer Meteorology and Pollution in the Western Sichuan Basin (1[st] BLMP - SCB) was carried out at a rural location (Sanbacun, 103°40′38″ E, 30°54′59″ N, 650 m) on the eastern foothills of the Tibetan Plateau from December 2018 to January 2019 (during the winter of 2018–2019), with a duration of approximately 40 days (Fig. 1). In this study, the vertical profiles of carbonaceous aerosols (black carbon [BC] and brown carbon [BrC]) measured by a micorAeth MA200, along with the horizontal wind profiles and vertical wind velocity ($w$) observed by a Doppler Wind Lidar mounted on a tethered balloon, were predominantly utilized. Moreover, all the

equipment employed and the corresponding parameters during the 1st BLMP - SCB campaign carried out at the rural site were presented in Table 1, as previously mentioned.

• Define the three groups (clusters???) early on in the article

**Response:** I express my sincere gratitude for your constructive suggestions regarding the improvement of the manuscript. As you have pointed out, it is of utmost significance to define the three clusters at the early stage of the article. Therefore, before presenting the results, the three typical types of secondary BrC (BrC$_{sec}$) profiles were defined in Table 2 of the subsection of "2.2.2 K-means cluster analysis" of "2.2 Methods" in the revised edition of our manuscript.

Table 2 Characteristics of the three clusters of BrC$_{sec}$ profiles.

| Cluster | Descriptions | Frequency |
| --- | --- | --- |
| 1 | Relatively uniform vertical distributions | 17.28% |
| 2 | Higher values at an altitude of 1.4 km above sea level (ASL) | 16.05% |
| 3 | More rapid decreases with the increase in altitude | 66.67% |

• Keep figure style consistent throughout to make comparisons easy

**Response:** I sincerely appreciate your reminder. As you have pointed out, maintaining consistent figure styles throughout the manuscript is of paramount importance for facilitating comparisons. Consequently, in the revised version of our manuscript, the styles of Figures 3 and 4, as well as Figures 8 and 9, have been standardized. Moreover, the styles of the figures in the supplementary information have also been made consistent with those in the main text to enhance the ease of comparison.

• The flow of the article is nonexistent. I do not have a good recommendation at the moment but the authors need to sit down and outline the article first and make sure the discussion and results come in both a logical and aesthetic way. The article is rather

lengthy and I feel like another pass at an outline will help the authors remove unnecessary information.

**Response:** I sincerely appreciate your constructive suggestions. As you have pointed out, the unclear flow of the article significantly undermines its readability. Consequently, the structure of the article has been reorganized. In fact, this study initially conducted an analysis of the vertical profiles of black carbon (BC) and brown carbon (BrC, classified into primary brown carbon ($BrC_{pri}$) and secondary brown carbon ($BrC_{sec}$) according to sources) in the eastern foothills of the TP. The concentrations of $BrC_{sec}$ were elevated in the upper atmosphere after noon, whereas the concentrations of BC and $BrC_{pri}$ were concentrated near the surface. More interestingly, it is discovered that subsequent to noon, the peak of the $BrC_{sec}$ profile progressively approaches the ground over time, whereas this phenomenon does not occur for BC and $BrC_{pri}$. Based on our prior research, mechanical turbulence triggered by wind shear on the leeward slope of the TP occurs frequently and is intense in the upper atmosphere. Consequently, it has been discovered that $BrC_{sec}$ in the upper atmosphere is transported downward into the PBL by the mechanical turbulence induced by wind shear. Furthermore, we also explored the influence of long - range and regional transport on the temporal variations of $BrC_{sec}$ profiles. The impacts were negligible as the gridded back - trajectory concentrations and pollution rose plots of BC and $BrC_{sec}$ were comparable.

• The results don't really show me how turbulence has any impact on the vertical profile of aerosols.

**Response:** I express my gratitude for your suggestions. As you have indicated, discerning the influence of turbulence on the vertical profiles of aerosols from the previous figures is not straightforward. Consequently, the relevant figures have been redrawn in the revised manuscript. As depicted in Fig. 2a, the peak of $BrC_{sec}$ profiles at 1.4 km ASL progressively approaches the ground from 11:00 to 20:00. Meanwhile, the ratio of $BrC_{sec}$ to BrC gradually moves away from the ground during the same time intervals. The variation in $BrC_{sec}$ profiles was synchronized with those of wind

shear ($S_{vec}$) and vertical wind velocity ($w$) (Fig. S2), suggesting a significant impact of the descending motion induced by wind shear on the $BrC_{sec}$ profiles.

[Figure]

Fig. 2 Diurnal variations of the vertical profiles of (a) $BrC_{sec}$, (b) $BrC_{pri}$, (c) $BrC_{sec}$/BrC ratio, and (d) BC during the campaign. These profiles represent the averages of all days throughout the campaign. The black arrows in (a) and (c) indicate the temporal movement of the peaks of profiles of $BrC_{sec}$ and the $BrC_{sec}$/BrC ratio.

[Figure]

Fig. S2 Diurnal variations in the vertical profiles of (a) vector shear ($S_{vec}$) and (b) vertical wind velocity ($w$) as observed by the Doppler Wind Lidar during the

campaign. These profiles represent the mean values across all days of the campaign. The white arrows denote the temporal progression of the peaks of profiles of $S_{vec}$ and $w$.

To more comprehensively elucidate the influence of mechanical turbulence on $BrC_{sec}$ profiles, the k - means cluster analysis method was employed to categorize $BrC_{sec}$ profiles into three clusters. Specifically, Cluster 1 comprises relatively uniform $BrC_{sec}$ profiles; Cluster 2 consists of $BrC_{sec}$ profiles featuring a peak at 1.4 km ASL; and Cluster 3 involves $BrC_{sec}$ profiles exhibiting a more rapid decline in $BrC_{sec}$ with increasing height (Fig. 6a). As is evident from Fig. 6, there exist substantial disparities in the profiles of environmental and meteorological variables across the three clusters. In Cluster 1, the $BrC_{sec}$ profile exhibited greater uniformity, with slightly elevated concentrations within the altitude range of 1.7 km to 2.1 km ASL. This phenomenon was closely associated with favorable atmospheric dispersion, specifically characterized by a more substantial temperature disparity between the lower and upper atmospheres (Fig. 6f), weak descending motion (Fig. 6g), and a uniform profile of turbulence intensity (Fig. 6j). In Cluster 2, the concentrations of $BrC_{sec}$ exhibited a notable peak at 1.4 km ASL. Moreover, the concentrations of $O_3$ were substantially higher, whereas those of $PM_1$ were comparatively lower when contrasted with those of Clusters 1 and 3. This finding implies that this cluster corresponds to secondary pollution. From the perspectives of meteorological factors, the temperature profiles of Cluster 2 and Cluster 3 were comparable. However, the descending motion and wind shear in the upper atmosphere were more pronounced in Cluster 2 than in Cluster 3. The turbulence intensity ($V_{TKE}$) exhibited comparable characteristics for Clusters 2 and 3, demonstrating a notable increase with the elevation from 1.5 km ASL. Nevertheless, the concentrations of secondary air pollutants ($BrC_{sec}$, $O_3$) below 2.0 km ASL in Cluster 3 were substantially lower than those in Cluster 2. Moreover, Cluster 1 predominantly emerges during the nighttime, whereas Cluster 2 mainly occurs during the daytime (Fig. 7a). Additionally, the occurrence frequency of Cluster 2 experiences a 23% increase from 3 to 15 hours following the occurrence of Cluster 1

(Fig. 7b). Consequently, it is deduced that the high concentrations of BrC$_{sec}$ at altitudes between 1.7 km and 2.1 km above sea level (ASL) during the nighttime were transported downward to the lower atmosphere during the daytime as the elevated turbulence induced by wind shear developed.

[Figure]

Fig. 6 (a) Three clusters of representative mean BrC$_{sec}$ profiles, and (b–e) the corresponding average vertical profiles of other air pollutants (BC, SO$_2$, O$_3$, and PM$_1$), and (f–j) the average vertical profiles of temperature (T), vertical wind velocity ($w$), vector shear ($S_{vec}$), wind - directional shear ($S_{dir}$), and turbulence intensity ($V_{TKE}$) during the field campaign.

[Figure]

Fig. 7 (a) Diurnal variations in the occurrence frequency of the three clusters of vertical profiles and (b–d) occurrence frequency of the other two clusters from 3 to 21 hours at three - hour intervals subsequent to the occurrence of the specific cluster (for instance, subplot (b) depicts the frequency of Clusters 2 and 3 from 3 to 21 hours following the occurrence of Cluster 1).

**Major Issues**

1. Figures (in general): The authors talk about daytime vs nighttime and impact of altitude on concentrations. However, all of the figures are very hard to interpret since there is no clear separation between day vs night and altitudes. Fig. S3 has some

altitude information but it is nigh on impossible to understand. I strongly recommend the authors organize their figures such that differences due to day, night, and altitude are very clear. Along with any other key parts the authors wish to highlight.

**Response:** Thank you for your good suggestions, which is very useful for improving the readability of the manuscript. As you mentioned, the previous figures are very hard to interpret due to no clear separation between day vs night and altitudes. In the revised manuscript and the supplemental information, the relevant figures (Figs. 3, 4, 5, and S3) were reorganized and clearly separated such that differences due to day, night, and altitude are very clear. In those figures, daytime (08:00, 11:00, 14:00, and 17:00, local time) and nighttime (02:00, 05:00, 20:00, and 23:00) conditions are indicated by transparent yellow and gray boxes, respectively. The size of the dot corresponds to altitude, with the altitude ranging from 0.65 to 2.70 km as the dot size increases.

2. *Line 146-149: "The performances of the…within PBL."* – this sentence is rather unclear. What sensors were validated? What were the reference instruments? A little bit more information here would be very helpful, yes there is a reference but checking a reference to understand the meaning of a sentence creates very poor readability.

**Response:** I sincerely appreciate your reminder. As you have pointed out, the sentence spanning Lines 146 to 149 exhibits a notable lack of clarity, encompassing the issues of the validated sensors and the reference instruments employed that remain unspecified. Consequently, the sentence has been revised to the blue - colored text in the revised version of our manuscript, as presented below.

The performance of the package (gaseous and particulate matter sensors) was validated through comparison with on - ground reference equipment (Model 49C, Model 42i, Model 450i, Model 48i - TLE, Model 5030i, Thermal Fisher, USA) with coefficients of determination ($R^2$) ranging from 0.81 to 0.93 (Pang et al., 2021). It was corroborated that the sensor package is a reliable device for aerial measurements of gaseous and particulate matter pollutants within the PBL.

3. Figure 1: - This figure is not very helpful to understand the experimental setup. A table describing the instrumentation might be more helpful. Further it is unclear from Fig. 1 if there are multiple aerosol samplers at different elevations along the Tibetan Plateau. I would recommend a 2-D map showing location of all sites and then a table describing the equipment at each site. Further since the authors are talking about gradients some information on the measurement heights is crucial.

**Response:** We express our gratitude for your valuable suggestions. The revised Figure 1 depicts the experimental configuration for two - dimensional pollution and meteorology within the PBL at the rural site of Sanbacun, located at the eastern foothills of the TP. The equipment utilized is presented and elaborated upon in Table 1. During the 1st BLMP - SCB, multiple aerosol samplers were deployed at different elevations along the Tibetan Plateau to collect $PM_1$ samples for the purpose of obtaining the chemical components in $PM_1$. Nevertheless, the relevant data was not analyzed in this study, and consequently, the relevant information was not included in Figure 1 and Table 1. The revised Figure 1 and added Table 1 have been provided in the response to the previous question.

4. Line 194-195: "…Hybrid Single-Particle Lagrangian Integrated Trajectory…" – what is this model? Did you make it or did it already exist? If it exists a reference would be needed. If you created it then more detail is necessary.

**Response:** Thank you for your reminder. In the revised version of our manuscript, more detailed information regarding the HYSPLIT model has been incorporated. The relevant content is presented in blue text as follows.

The Hybrid Single - Particle Lagrangian Integrated Trajectory (HYSPLIT) model, developed by the NOAA Air Resource Laboratory (Draxler and Hess, 1998), has been widely employed for the investigation of atmospheric transport and dispersion.

5. Line 203: "…combining BC and UVPM…" – I'm still not entirely sure what

UVPM is as it was poorly described in the preceding sections.

**Response:** Thank you for catching that. Actually, UVPM is brown carbon (BrC), and thus it was replaced with BrC throughout the manuscript. Brown carbon was from primary sources and secondary formation, which was marked as $BrC_{pri}$ and $BrC_{sec}$, respectively. The BrC, $BrC_{pri}$, and $BrC_{sec}$ were defined when they first appears in the revised manuscript.

6. Section 2.3 – A more thorough description here would be very helpful, perhaps with a figure. You say you used cluster analysis to divide the UVPM profiles in to three groups– this would be a great section in which to introduce those three groups with a table so that the reader knows what to expect in the latter sections.

**Response:** We sincerely appreciate your constructive suggestions, which are highly valuable for enhancing the manuscript. In the revised manuscript, the section titled "2.3 K - means cluster analysis" will be described in greater detail, as presented in the blue text below.

The determination of the number of clusters is crucial for attaining novel and accurate understandings. Drawing upon the Elbow Method (Syakur et al., 2018) and the characteristic profiles of air pollutants, k - means clustering analysis was employed to partition the $BrC_{sec}$ profiles during the campaign into three clusters, with each cluster featuring a comparable vertical structure of $BrC_{sec}$. The characteristics of each cluster of the BrC profile are presented in Table 2. The k - means clustering algorithm provided by MATLAB$^{©}$ was employed.

Table 2 Characteristics of the three clusters of $BrC_{sec}$ profiles.

| Cluster | Descriptions | Frequency |
| --- | --- | --- |
| 1 | Relatively uniform vertical distributions | 17.28% |
| 2 | Higher values at an altitude of 1.4 km above sea level (ASL) | 16.05% |

| 3 | More rapid decreases with the increase in altitude | 66.67% |
| --- | --- | --- |

7. Section 2.4: - This section seems like it needs to be higher up – see my note about organizing this entire section under "General Comments"

**Response:** Thank you for your suggestions. In the revised manuscript, Section 2.4 from the previous manuscript will be relocated upward to "2.1.2 Data collection and quality control". Moreover, the relevant contents will be substantially enhanced as presented in the following blue text.

......

The micorAeth MA200 (AethLabs, USA) was employed to quantify the mass concentrations of carbonaceous constituents of aerosol particles across five wavebands (375 nm, 470 nm, 528 nm, 625 nm, and 880 nm). The carbonaceous particles measured at 880 nm by the instrument were typically construed as BC, whereas those in the 375 nm ultraviolet band were regarded as BrC, which is either emitted from primary sources ($BrC_{pri}$) or produced through secondary reactions ($BrC_{sec}$). The MA200 aspirates an air sample at a flow rate of 100 ml min$^{-1}$ through a 3 - mm - diameter portion of the filter media. The light attenuation (ATN) in response to absorbance of particles collected on the 'Sensing' spot is measured relative to an adjacent 'Reference' portion of the filter where no particles are accumulated ($\Delta$ATN). A temporal resolution of 5 seconds ($\Delta$t) was established to align with other observations conducted during the campaign. Carbonaceous particles exhibit a proportional relationship with the rate of change of the ATN ($\Delta$ATN/$\Delta$t). A challenge posed by high time resolution measurements is that, even when high deposition rates of absorbing materials are employed, within short timebases, $\Delta$ATN can be small enough to be notably affected by measurement noise.

The micorAeth MA200 might yield negative values under conditions of lower mass concentrations and higher temporal resolution, which can account for up to 30% of

the uncertainty associated with the filter - based optical attenuation technique (Hagler et al., 2011). Consequently, the raw data acquired for vertical profiles must be rectified prior to analyzing their characteristics, particularly for in - situ observations at high altitudes. The mere removal of negative values is an inappropriate approach, as it would neglect the corresponding positive fluctuations caused by noise and lead to an upward bias in the final data. Averaging data over an extended time frame typically mitigates the noise within the signal; however, this may conflict with the requirement for high temporal - resolution data. Post - processing strategies such as moving averages or advanced mathematical techniques can be utilized to isolate the noise and reconstruct the time series (Kostelich and Schreiber, 1993). Nevertheless, these methods fail to leverage the ATN values associated with the internal load rate of the filter and the knowledge regarding the successive difference characteristic of the MA200. The optical noise - reduction averaging (ONA) algorithm, devised by Hagler et al. (2011), aims to perform adaptive time - averaging of carbonaceous components so as to mitigate the noise in BC data.

The ONA algorithm conducts smoothing processing on the time series of carbonaceous particles via a user - specified minimum attenuation change ($\Delta ATN_{min}$). For a given concentration of carbonaceous aerosols, this process leads to an adjusted timebase ($\Delta t'$). When the concentration reaches a sufficiently high level or the intrinsic timebase is long, $\Delta ATN$ will exceed $\Delta ATN_{min,}$ and the intrinsic time resolution will be maintained. Nevertheless, in the case of relatively low concentrations or short timebases, $\Delta ATN$ will be lower than $\Delta ATN_{min,}$ and the time series will be smoothed over the time intervals $\Delta t' > \Delta t$ required to attain $\Delta ATN_{min}$. A second constraint is that the ATN value at the conclusion of the interval $\Delta t'$ must be the final instance of that value within the remaining part of the time - series for that specific sample spot. Consequently, $\Delta t_i'$ is extended to the ultimate occurrence of that ATN value. The frequency of negative values is reduced by applying the constraint. This is because when the ATN value returns to the same level later in the time - series, it implies that $\Delta ATN < 0$ at that specific time step, which leads to a negative

concentration. A consequence of the second constraint is that a certain level of smoothing exists even when $\Delta ATN_{min}$ equals zero. In principle, the average concentration of carbonaceous particles during the time interval $\Delta t_i$' can be calculated using $\Delta ATN_i/\Delta t_i$'. Nevertheless, the duration of light transmission measurement at a high temporal resolution is relatively brief, rendering it prone to noise interference. The mean concentration of carbonaceous particles during the time interval $\Delta t_i$' is calculated by averaging the set of concentrations reported at the intrinsic timebase within that interval. The incidence of negative concentrations should be less than 30% of the datasets to guarantee data quality. This is a straightforward method for addressing the noise in real - time data from the micorAeth MA200. It achieves this by dynamically adjusting the competing factors of averaging time and noise, thereby preserving the highest possible temporal resolution within the datasets. The algorithm leads to substantial reductions in noise and considerably more rational temporal variations in the mass concentrations of carbonaceous particles (Cheng and Lin, 2013; Park et al., 2010). The program was employed to conduct post - processing on the negative values obtained from our real - time profile measurements.

The estimation of secondary brown carbon ($BrC_{sec}$) holds significant importance in ascertaining the proportion of $BrC_{sec}$ within BrC. Initially, the minimum ratio of BrC to BC, denoted as $(BrC/BC)_{min}$, was employed as a surrogate for the ratio of $BrC_{pri}$ to BrC ($BrC_{pri}/BrC$) to estimate the mass concentrations of $BrC_{sec}$ (Castro et al., 1999). Nevertheless, numerous studies have indicated that the $(BrC/BC)_{min}$ demonstrates a certain level of randomness in actual observations, which results in substantial errors, particularly for the low BC concentrations in high - altitude regions. To tackle this issue, Lim and Turpin (2002) put forward the approach of arranging the BrC/BC ratios in ascending order and substituting the $BrC_{pri}/BC$ ratio with the mean value of the top 10%–20% of the data. However, there is a dearth of a universally applicable criterion for determining the appropriate percentile range. In light of the disparate sources of $BrC_{sec}$ and BC, Millet et al. (2005) put forward a method for estimating $BrC_{sec}$ concentrations by utilizing the minimum correlation coefficient between BrC

and BC. This methodology aims to determine the $BrC_{pri}/BC$ ratio (designated as $(BrC/BC)_{pri}$) at which the correlation between $BrC_{sec}$ and BC reaches its minimum, and this ratio is employed as the $BrC_{pri}/BC$ ratio. Adopting this approach, Wu and Yu (2016) devised a toolkit within Igor Pro for calculating the mass concentration of $BrC_{sec}$. This development notably improved the precision of $BrC_{sec}$ estimation, as presented in Eqs. (1) and (2).

$$BrC_{pri} = (BrC/BC)_{pri} \times BC, \qquad (1)$$

$$BrC_{sec} = BrC - BrC_{pri}. \qquad (2)$$

In Eq. (1), $(BrC/BC)_{pri}$ denotes the ratio of the concentrations of $BrC_{pri}$ to BC during the campaign. Based on the measurements of BrC and BC, $BrC_{pri}$ and $BrC_{sec}$ can be estimated by means of Eqs. (1) and (2).

......

8. Line 237: - UVPM Ratio is undefined

**Response:** We express our gratitude for your identification of this issue. The original expression was inaccurate and has been rectified to "the ratio of $BrC_{pri}$ to BrC ($BrC_{pri}/BrC$)" in the revised version of our manuscript.

9. Line 236: "…the fraction of UVPMsec in UVPM." – UVPM is very poorly defined, I thought all UVPM was 'secondary'? More care needs to be taken to describe these particle types in the introduction and data collection sections.

**Response:** We express our gratitude for your inquiries and suggestions. In fact, based on the optical properties of aerosol particles, UVPM is identified as brown carbon (BrC). Brown carbon is produced via the inefficient combustion of hydrocarbons and the photo - oxidation of biogenic particles (Andreae and Gelencser, 2006), which are respectively referred to as primary sources ($BrC_{pri}$) and secondary formation ($BrC_{sec}$). In the revised version of our manuscript, the BrC particles are more comprehensively described in the introduction and data collection sections.

10. Equation 1 & 2: It is now very unclear to me now what aerosol types you are measuring and which types you are estimating through relationships. Further now there is a UVPMpri designation which is not defined (first shows up on Line 237).

**Response:** Thank you for catching that. The micorAeth MA200 (AethLabs, USA) was employed to quantify the mass concentrations of carbonaceous constituents of aerosol particles across five wavebands (375 nm, 470 nm, 528 nm, 625 nm, and 880 nm). The carbonaceous particles measured at 880 nm by the instrument were typically construed as BC, whereas those in the 375 nm ultraviolet band were regarded as BrC, which is either emitted from primary sources ($BrC_{pri}$) or produced through secondary reactions ($BrC_{sec}$). The $BrC_{pri}$ and $BrC_{sec}$ can be estimated through certain relationships. The revised text is showed with the blue text as follows.

The estimation of secondary brown carbon ($BrC_{sec)}$ holds significant importance in ascertaining the proportion of $BrC_{sec}$ within BrC. Initially, the minimum ratio of BrC to BC, denoted as $(BrC/BC)_{min}$, was employed as a surrogate for the ratio of $BrC_{pri}$ to BrC ($BrC_{pri}/BrC$) to estimate the mass concentrations of $BrC_{sec}$ (Castro et al., 1999). Nevertheless, numerous studies have indicated that the $(BrC/BC)_{min}$ demonstrates a certain level of randomness in actual observations, which results in substantial errors, particularly for the low BC concentrations in high - altitude regions. To tackle this issue, Lim and Turpin (2002) put forward the approach of arranging the BrC/BC ratios in ascending order and substituting the $BrC_{pri}/BC$ ratio with the mean value of the top 10%–20% of the data. However, there is a dearth of a universally applicable criterion for determining the appropriate percentile range. In light of the disparate sources of $BrC_{sec}$ and BC, Millet et al. (2005) put forward a method for estimating $BrC_{sec}$ concentrations by utilizing the minimum correlation coefficient between BrC and BC. This methodology aims to determine the $BrC_{pri}/BC$ ratio (designated as $(BrC/BC)_{pri}$) at which the correlation between $BrC_{sec}$ and BC reaches its minimum, and this ratio is employed as the $BrC_{pri}/BC$ ratio. Adopting this approach, Wu and Yu (2016) devised a toolkit within Igor Pro for calculating the mass concentration of $BrC_{sec}$. This development notably improved the precision of $BrC_{sec}$ estimation, as

presented in Eqs. (1) and (2).

$$BrC_{pri} = (BrC/BC)_{pri} \times BC, \qquad (1)$$

$$BrC_{sec} = BrC - BrC_{pri}. \qquad (2)$$

In Eq. (1), $(BrC/BC)_{pri}$ denotes the ratio of the concentrations of $BrC_{pri}$ to BC during the campaign. Based on the measurements of BrC and BC, $BrC_{pri}$ and $BrC_{sec}$ can be estimated by means of Eqs. (1) and (2).

11. *Equation 3:* Is "mechanical turbulence index" a term the authors came up with? This is the square root of the turbulent kinetic energy (TKE) which is like some root mean square velocity. Further TKE encompasses all turbulence and not just shear which is what the authors are trying to isolate. I strongly recommend the authors look at the TKE budget equation and go from there.

**Response:** I express my sincere gratitude for your constructive suggestions, which have notably enhanced the readability of the manuscript. In accordance with the research findings of Lan et al. (2018) and Sun et al. (2012), the term "mechanical turbulence index" has been modified to "turbulent velocity scale" in the revised manuscript. As you suggested, The TKE budget equation offers a more comprehensive and in - depth understanding of the generation, transport, and dissipation of turbulence along a specific transect. In the context of unstable stratification, a significant correlation exists between TKE and the effects of mechanical shear and buoyancy (Yue et al., 2015). During the campaign, the buoyancy effect is notably suppressed on cloudy days (Song and Zhang, 2024). Furthermore, considering the constraints of the deployed instrumentation and the available measurements with relatively low temporal resolution, this study only analyzed the mechanical shear (including wind speed shear and directional shear). the revised contents were showed with the blue text as follows.

Turbulent kinetic energy (TKE) serves as an indicator of turbulence intensity, and it represents a crucial variable for the comprehension of the exchanges of energy, water

vapor, and greenhouse gases between the land surface and the atmosphere. The turbulent velocity scale, defined as $V_{TKE}=\left[(1/2)\left(\overline{u'^2}+\overline{v'^2}+\overline{w'^2}\right)\right]^{1/2}=\sqrt{TKE}$, has been widely employed to characterize turbulence intensity (Lan et al., 2018; Sun et al., 2012). Here, $u$, $v$, and $w$ denote the zonal, meridional, and vertical wind components respectively, while $u'$, $v'$, and $w'$ signify the standard deviation of each variable. The vertical profiles of $V_{TKE}$ can be derived from the aforementioned equation. A positive correlation exists between the $V_{TKE}$ and mechanical turbulence, such that an increase in $V_{TKE}$ corresponds to an enhancement of mechanical turbulence. The combination of the profiles of $V_{TKE}$ and air pollutants can be employed to gain a more comprehensive understanding of the downward transport of air pollutants at the eastern foothills of the Tibetan Plateau.

The TKE budget equation offers a more comprehensive and in - depth understanding of the generation, transport, and dissipation of turbulence along a specific transect (Martilli et al., 2002, Eq. 4).

$$\frac{1}{2}\frac{\partial e^2}{\partial t}=-\overline{u'w'}\frac{\partial \overline{u}}{\partial z}+\frac{g}{T}\left(\overline{w'\theta'}\right)-\frac{1}{2\rho}\frac{\partial \overline{\left(w'e^2\right)}}{\partial z}-\frac{1}{\rho}\frac{\partial \overline{w'p'}}{\partial z}-\nu\left(\overline{\frac{\partial u_i'}{\partial x_j}\frac{\partial u_i'}{\partial x_j}}\right), \qquad (4)$$

where $e$ represents the TKE, $t$ denotes time, $g$ signifies the acceleration due to gravity, $T$ stands for temperature, $\theta$ represents the potential temperature, $\overline{w'\theta'}$ refers to the turbulent heat flux, $\overline{u'w'}$ indicates the turbulent momentum, $\rho$ denotes the air density, $u$ represents the wind speed, $z$ represents the observational height, $w'$ represents the vertical wind velocity fluctuation, and $p'$ represents the atmospheric pressure pulsation. The value of TKE is primarily influenced by mechanical shear and buoyancy effects (represented by the first two terms on the right - hand side of Equation 4). In the context of unstable stratification, a significant correlation exists between TKE and the effects of mechanical shear and buoyancy (Yue et al., 2015). During the campaign, the buoyancy effect is notably suppressed on cloudy days (Song and Zhang, 2024). Furthermore, considering the constraints of the deployed instrumentation and the available measurements with relatively low temporal

resolution, this study only analyzed the mechanical shear (including wind speed shear and directional shear).

12. Equation 4: This notation is very unclear. Shear is a derivative so should look something like $\frac{du}{dz}$ where $u$ is the streamwise wind speed.

**Response:** Thank you for your reminder. As you pointed out, the notation in Equation 4 is highly ambiguous. Consequently, the equation has been revised, and Fig. S1 has been incorporated into the supplementary information to enhance the clarity of the expression. The revised contents and the added Fig. S1 are presented as follows.

According to Mahrt (2017), the wind speed shear was defined as

$$Sh \equiv \delta V = \left|V_{(50+i)\,m}\right| - \left|V_{i\,m}\right|, \qquad\qquad (5)$$

where $\delta V$ denotes the disparities in wind speed ($V$) between adjacent measurement levels with an interval of 50 meters (Fig. S1), and $i$ represents the height at the specific level.

[Figure]

Fig. S1 The wind direction terminology encompasses the vector shear (represented by the green symbol), the negative of the vector shear (depicted by the rhombus head), and the speed shear (indicated by the black symbol). The symbols $V (i + 50$ m) and $V (i$ m) denote the wind vectors at the altitudes of $(i + 50)$ m and $i$ m, respectively.

13. Equation 5: I do not understand what this equation gives. How are you defining your streamwise (u) and spanwise (v) velocities? Traditionally $\bar{v} = 0$ since the coordinates are rotated such that the mean wind speed $U = \bar{u}$ and $\bar{v} = \bar{w} = 0$ where $\bar{u}$ is the streamwise wind velocity, $\bar{v}$ is the spanwise wind velocity, and $\bar{w}$ is the vertical wind velocity.

**Response:** Thank you for catching that. I am very sorry due to unclear expression in the previous manuscript. The revision was showed with the blue-colored text as follows.

Furthermore, $S_{vec}$ was defined as the magnitude of the vector shear, which was determined by the vertical disparities in the wind - speed components.

$$S_{vec} \equiv \delta V = \sqrt{(\delta u)^2 + (\delta v)^2} = \sqrt{\left(\left|u_{(50+i)\,m}\right| - \left|u_{i\,m}\right|\right)^2 + \left(\left|v_{(50+i)\,m}\right| - \left|v_{i\,m}\right|\right)^2}, \qquad (6)$$

where $u$ and $v$ denote the zonal and meridional wind components, respectively. The wind - directional shear can be quantitatively characterized as the disparity between the magnitude of the vector shear and the speed shear.

$$S_{dir} \equiv S_{vec} - Sh, \qquad\qquad\qquad (7)$$

where $S_{dir}$ is denoted in m s$^{-1}$.

14. Figure 2: Text is very small and colors are random. In an earlier section the authors mentioned a difference in profiles between day and night. It would make sense to separate these figures based on day and night. Having the colors be a gradient from start time to end time would greatly improve readability. The authors could also consider a height time concentration plot. See example below except instead of particle size on the y-axis you will have your elevation above ground. It is also unclear how these profiles were taken, was the tethered balloon raised to 2.5 km above ground level and back down at each time stamp? There are no labels on the panels of this figure, and the caption does a poor job of describing what the reader is looking at. I've also included a caption below the example figure so the authors can understand what a caption should include

**Response:** We express our sincere gratitude for your constructive recommendations, which are of great significance in enhancing the readability of all figures. We offer our apologies for any inconvenience that may have arisen. In accordance with your suggestions and the example figure provided, Figure 2 has been replaced with a height-time-concentration plot featuring enlarged text and gradient colors. The diurnal variations in the vertical profiles of vector shear ($S_{vec}$) and vertical wind velocity ($w$) as observed by the Doppler Wind Lidar during the campaign were showed in Fig. S2. In the two figures, those profiles represent the averages of all days throughout the campaign. The black and white arrows indicate the temporal movement of the peaks of $BrC_{sec}$, the $BrC_{sec}/BrC$ ratio, $S_{vec}$, and $w$ at a specific altitude.

The tethered balloon ascends or descends at a constant velocity of 0.5 m s$^{-1}$. It is regulated by an electric winch from the ground to reach a specific altitude. When the wind velocity in the upper atmosphere exceeds 5.0 m s$^{-1}$, the balloon is retracted to guarantee the safety of the onboard instruments. The variables were monitored at three - hour intervals (02:00, 05:00, 08:00, 11:00, 14:00, 17:00, 20:00, 23:00) as the balloon ascended and descended at each timestamp during the campaign. Each upward and downward sounding endures approximately two hours. The variation in conditions between upward and downward sounding is not substantial. Consequently, the profile at each timestamp represents an average of the upward and downward sounding. The aforementioned explanations, along with the revised Figure 2 and Figure S2, will be incorporated into the revised manuscript or supplementary information.

[Figure]

Fig. 2 Diurnal variations of the vertical profiles of (a) BrC$_{sec}$, (b) BrC$_{pri}$, (c) BrC$_{sec}$/BrC ratio, and (d) BC during the campaign. These profiles represent the averages of all days throughout the campaign. The black arrows in (a) and (c) indicate the temporal movement of the peaks of profiles of BrC$_{sec}$ and the BrC$_{sec}$/BrC ratio.

[Figure]

Fig. S2 Diurnal variations in the vertical profiles of (a) vector shear ($S_{vec}$) and (b) vertical wind velocity (*w*) as observed by the Doppler Wind Lidar during the

campaign. These profiles represent the mean values across all days of the campaign. The white arrows denote the temporal progression of the peaks of profiles of $S_{vec}$ and $w$.

15. Lines 298-305: "The vertical distribution of…discussed in details in the following sections." –Nearly everything that was written in these lines is impossible to read from Fig. 2.

**Response:** Thank you for identifying that issue. In accordance with the revised Figure 2 and Figure S2, the sentences spanning Lines 298 - 305 have been revised to the text marked in blue, as presented below.

Compared with profiles of $BrC_{pri}$, the profiles of $BrC_{sec}$ were more uniform, and the differences among the $BrC_{sec}$ profiles were more significant than those among the $BrC_{pri}$ profiles at the different time points. Regarding the $BrC_{sec}/BrC$ ratio, the peak of the profile progressively distanced itself from the ground between 11:00 and 20:00 (Fig. 2c), which was closely associated with the contrasting variations of $BrC_{sec}$ and $BrC_{pri}$. More interestingly, the peak of $BrC_{sec}$ profiles at 1.4 km ASL progressively approached the ground from 11:00 to 20:00 (Fig. 2a). The variation in $BrC_{sec}$ profiles was synchronized with those of wind shear ($S_{vec}$) and vertical wind velocity ($w$) (Fig. S2), suggesting a significant impact of the descending motion induced by wind shear on the $BrC_{sec}$ profiles.

16. Figure S1 & Figure 3: S1 seems like it would be more useful in the paper instead of in supplemental information. Further, the way both Figure 3 and S1 are presented makes it hard to see any trend. S1 is easier since BC and $UVPM_{pri}$ basically follow each for all time periods but then in Fig. 3 it is unclear if ever the two aerosol types follow each other. Again it would be helpful to separate these plots by day and night since that is what the paper talks about.

**Response:** Thank you for your constructive suggestions. Figure S1 was moved to the main text and renamed as Figure 3, and the previous Figure 3 was renamed as Figure

4. Furthermore, the two figures were reorganized to separate those subplots by day vs night and altitude. The situations during daytime (08:00, 11:00, 14:00, and 17:00, local time) and nighttime (02:00, 05:00, 20:00, and 23:00) were delineated by transparent yellow and gray boxes, respectively. The size of the squares signifies altitude, with the altitude ranging from 0.65 to 2.70 km as the squares increase in size.

The concentrations of $BrC_{sec}$ exhibited a nonlinear variation in relation to BC across different time of the day (Fig. 4). During the daytime, the $BrC_{sec}$ exhibited an increasing trend with the rise in BC concentrations. Subsequently, as the BC concentrations continued to increase, the $BrC_{sec}$ less varied. The concurrent increases in $BrC_{sec}$ and BC suggest that low concentrations of primary emissions are conducive to secondary formation. In contrast, a higher quantity of primary particles inhibits secondary formation through a series of processes, including the coagulation of newly - formed particles by large particles and the scattering of solar radiation. During the nocturnal period, the correlation between secondary $BrC_{sec}$ and BC is not statistically significant. However, a pronounced synchronous increase in $BrC_{sec}$ and BC is evident in the upper atmosphere (represented by the larger squares), suggesting that primary emissions in the upper air facilitate secondary formation during the night.

[Figure]

Fig. 4 Diurnal variations in the relationships between the profiles of BC and $BrC_{sec}$

during the campaign. The relationships were fitted using binary linear regression (represented by the black lines), and the coefficients of determination ($R^2$) were presented in each sub - plot. The situations during daytime (08:00, 11:00, 14:00, and 17:00, local time) and nighttime (02:00, 05:00, 20:00, and 23:00) were respectively delineated by transparent yellow and gray boxes. The size of the hollow square signifies altitude, with the altitude ranging from 0.65 to 2.70 km as the square increases in size. The $R^2$ followed by double-asterisk and single-asterisk respectively achieved statistical significance at the 0.01 and 0.05 levels.

17. Figure 4: It is unclear what I am looking at here. No labels are present. It is also unclear how these calculations were done. I'm sure it was a mix of the model from earlier and whatever "CWT" is. It is not clear at all how the differences between BC and UVPM source areas is calculated.

**Response:** We appreciate your questions and suggestions. The primary reason for the unclear expression is the absence of an introduction to the CWT method. In the revised manuscript, we have introduced this method in the section titled "2.2.1 Identification of potential source regions for BC and BrC$_{sec}$," which is presented in blue text as follows. According to your suggestions, in the revised Fig. 11, UVPM source areas is revised as BrC$_{sec}$ source areas. In the figure, the black solid dot indicates the location of the rural site (Sanbacun). The light blue dotted lines in each sub - plot represent longitude and latitude. The disparities between the source areas of BC and BrC$_{sec}$ are calculated by subtracting the gridded back - trajectory concentrations of BrC$_{sec}$ from the gridded back - trajectory concentrations of BC.

Based on the trajectories, the concentration - weighted trajectory (CWT) method (Hsu et al., 2003) was employed to identify the potential source regions of BC and BrC$_{sec}$. In the CWT method, each grid cell is assigned a weighted concentration through the calculation of the average of the sample pollutant concentrations. These sample pollutant concentrations are associated with trajectories that cross the grid cell, and the process is as follows:

$$C_{ij} = \frac{1}{\sum_{l=1}^{M} \tau_{ijl}} \sum_{l=1}^{M} C_l \, \tau_{ijl}, \qquad\qquad (3)$$

where $C_{ij}$ denotes the average weighted concentration within the ij$^{th}$ cell, $l$ represents the index of the trajectory, $C_l$ signifies the pollutant concentration measured upon the arrival of trajectory $l$, $M$ stands for the total quantity of trajectories, and $\tau_{ijl}$ indicates the time elapsed by trajectory $l$ within the ij$^{th}$ cell. A high $C_{ij}$ value signified that air parcels traversing the ij$^{th}$ cell would be correlated with a high concentration of pollutants at the receptor site.

[Figure]

Fig. 11 Gridded back - trajectory concentrations, which illustrate the mean concentrations of BC and BrC$_{sec}$, concentrations are derived using the CWT approach at heights of 100 m, 700 m, and 1300 m AGL during the campaign. The CWT method was presented in the section of "2.2 Methods". The disparities between the source areas of BC and BrC$_{sec}$ are calculated by subtracting the gridded back - trajectory concentrations of BrC$_{sec}$ from the gridded back - trajectory concentrations of BC. The black solid dot indicates the location of the rural site (Sanbacun). The light blue dotted lines in each sub - plot represent longitude and latitude.

18. Figure 5: UVPM and BC concentrations are flipped with respect to the previous plot making it hard to make inter-plot comparisons. Labels are very hard to read from a comfortable reading distance. It is not clear where the data for this plot comes from.

**Response:** Thanks for identifying that issue. In accordance with your suggestion, the concentrations of $BrC_{sec}$ and BC have been maintained in consistency with Figure 11, facilitating inter - plot comparisons. Moreover, the labels have been enlarged to enhance readability. The data utilized encompasses the mass concentrations of BC and $BrC_{sec}$, as well as wind data. The concentrations of BC were measured online via a micorAeth MA200, and the concentrations of $BrC_{sec}$ were calculated using Equations (1–2). The wind speed and direction data were acquired from the measurements of a Doppler Wind Lidar at Sanbacun. The aforementioned explanations have been incorporated into the caption.

[Figure]

Fig. 10 Pollution rose plots of BC and $BrC_{sec}$ at heights of 100 m, 700 m, and 1300 m AGL at the rural site (Sanbacun) during the campaign.The mean mass concentrations of BC and $BrC_{sec}$ and calm frequency were also presented with green-colored text in the corresponding sub - plot. The concentrations of BC measured online using a micorAeth MA200, and concentrations of $BrC_{sec}$ were calculated using Eq. (1–2). The wind speed and direction data were obtained from the observations of a Doppler Wind

Lidar at Sanbacun.

19. Line 395-396: "…three groups…" – These groups are not defined at all. Because they are not defined it is very hard to follow the remainder of this section.

**Response:** Thank you for your reminder. In the revised manuscript, the early section titled "2.3 K - means cluster analysis" defined the three clusters, as presented in the blue text and Table 2 below.

The determination of the number of clusters is crucial for attaining novel and accurate understandings. Drawing upon the Elbow Method (Syakur et al., 2018) and the characteristic profiles of air pollutants, k - means clustering analysis was employed to partition the $BrC_{sec}$ profiles during the campaign into three clusters, with each cluster featuring a comparable vertical structure of $BrC_{sec}$. The characteristics of each cluster of the $BrC_{sec}$ profile are presented in Table 2. The k - means clustering algorithm provided by MATLAB© was employed.

Table 2 Characteristics of the three clusters of $BrC_{sec}$ profiles.

| Cluster | Descriptions | Frequency |
|---|---|---|
| 1 | Relatively uniform vertical distributions | 17.28% |
| 2 | Higher values at an altitude of 1.4 km above sea level (ASL) | 16.05% |
| 3 | More rapid decreases with the increase in altitude | 66.67% |

20. *Figure 6:* Because the turbulence terms are defined in a way that I do not understand I do not know how to interpret this plot. Why are error bars all of the sudden included here and not on previous plots? I also don't see significant differences in the three clusters except in $UVPM_{sec}$ concentrations.

**Response:** Thank you for your questions and suggestions. The turbulent velocity scale, defined as $V_{TKE} = \left[ (1/2)\left( \overline{u'^2} + \overline{v'^2} + \overline{w'^2} \right) \right]^{1/2} = \sqrt{TKE}$, has been widely employed to

characterize turbulence intensity (Lan et al., 2018; Sun et al., 2012). Here, $u$, $v$, and $w$ denote the zonal, meridional, and vertical wind components respectively, while $u'$, $v'$, and $w'$ signify the standard deviation of each variable. The detailed explanations were given in the section of "2.2.3 Utilization of turbulence parameters" in the revised manuscript. In the revised Fig. 6, the error bars in the sub - plots were eliminated to maintain consistency with the preceding figures. Detailed explanations regarding Fig. 5 are provided in the following blue text.

In Fig. 6, notable disparities are present in the profile structure and magnitude of $BrC_{sec}$ among the three clusters. Cluster 1, which accounts for 17.28% of all profiles, exhibits weak fluctuations in the mass concentrations of $BrC_{sec}$, BC, and $PM_1$ in the vertical direction, with a weak peak of $BrC_{sec}$ at approximately 2.0 km ASL. In comparison with Clusters 2 and 3, the atmospheric stratification exhibits greater instability due to the significantly larger temperature disparity between the low - level and upper air (Fig. 6f), which results in a more robust ascending motion below 2.0 km above sea level (ASL). In comparison with Clusters 2 and 3, the vertical wind shear ($S_{vec}$, $S_{dir}$) and mechanical turbulence ($V_{TKE}$) are relatively weaker, which might be primarily associated with the occurrence of Cluster 1 during the nighttime. During the nighttime, $O_3$ concentrations remained at a low level as a result of the feeble photochemical reactions. Consequently, the more homogeneous $BrC_{sec}$ profiles during the nocturnal period are primarily regulated by thermodynamic processes (temperature).

Cluster 2, with a frequency comparable to that of Cluster 1 (16.05%), exhibits the mildest primary particulate matter (PM) pollution (BC, $PM_1$). Conversely, the $BrC_{sec}$ below 1.7 km above sea level (ASL) (Fig. 6a) and ozone ($O_3$) across the entire layer (Fig. 6d) display the most severe pollution levels among all clusters. In comparison to Cluster 1, the concentration of $BrC_{sec}$ below 1.7 km ASL is significantly higher, exhibiting a distinct peak at approximately 1.4 km ASL. Conversely, above this altitude, it rapidly declines to below 0.3 $\mu g\ m^{-1}$ at around 2.5 km ASL, which

represents the lowest value among the clusters. From the perspectives of meteorological factors, the temperature is significantly lower than that of Cluster 1 and comparable to that of Cluster 3 throughout the entire layer. In contrast to Cluster 1, the subsiding motion persists throughout the entire layer, reaching its maximum intensity at an altitude of 2.0 – 2.5 km ASL. Moreover, wind shear (represented by $S_{vec}$ and $S_{dir}$) and turbulence intensity ($V_{TKE}$) exhibit significantly higher magnitudes in the upper air. This phenomenon may be intricately associated with its occurrence during the daytime. Consequently, by integrating the vertical profiles of primary and secondary pollutants with meteorological factors, it is deduced that the rapid decline of $BrC_{sec}$ with the increasing altitude for Cluster 2 is primarily governed by dynamic processes (wind and turbulence).

Cluster 3, which constitutes two - thirds of the profiles, is the most prevalent during the campaign (66.67%). The cluster exhibits a uniform distribution throughout the day, with a range of 9% to 15%. Within the cluster, the concentrations of $BrC_{sec}$ below 2.0 km ASL and $SO_2$ across the entire layer are the lowest among the three clusters. Analogous to Cluster 2, there exists a weak ascending motion below 1.0 km ASL, which gradually transitions to a subsiding motion as the altitude increases, attaining the maximum intensity at 2.0 km ASL. Moreover, the vertical structure and magnitude of the $V_{TKE}$ are also comparable to those of Cluster 2. The dynamic processes (downward motion and mechanical turbulence) exhibit comparability between Clusters 2 and 3. However, the vertical profile of $BrC_{sec}$ is more uniform in Cluster 3, which can be attributed to the relatively lower concentration of $BrC_{sec}$ in the upper atmosphere, namely, the scarcity of material sources. Consequently, Cluster 3 characterizes the background profile of $BrC_{sec}$ at the observation site throughout the campaign.

[Figure]

Fig. 6 (a) Three clusters of representative mean $BrC_{sec}$ profiles, and (b–e) the corresponding average vertical profiles of other air pollutants (BC, $SO_2$, $O_3$, and $PM_1$), and (f–j) the average vertical profiles of temperature (T), vertical wind velocity ($w$), vector shear ($S_{vec}$), wind - directional shear ($S_{dir}$), and turbulence intensity ($V_{TKE}$) during the field campaign.

21. *Figure 8:* See point for Figure 2

**Response:** Thank you for your suggestion. Referring to your suggestions on Figure 2, The previous Figure 8 was changed to a time-height-concentration plot. The black arrows in the corresponding subplots indicate the temporal movement of the peaks of $BrC_{sec}$, Sh, and $S_{vec}$ at a specific altitude.

[Figure]

Fig. 8 Diurnal variations in vertical profiles of air pollutants (BrC$_{sec}$, BC, BrC$_{sec}$/BrC ratio, and PM$_1$) and meteorological factors (temperature, $w$, Sh, $S_{vec}$, and $S_{dir}$) on 7 January 2019. The black arrows indicate the temporal movement of the peaks of profiles of BrC$_{sec}$, Sh, and $S_{vec}$.

22. *Conclusion:* This section is way too vague. "novel findings" may be appropriate for a conference abstract but not so much for an article. The conclusion does a poor job in tying together the entire paper.

**Response:** I express my sincere gratitude for your valuable suggestions. In the revised manuscript, the conclusion section, which ties together the entire paper, has undergone substantial revision. The revised content is presented in blue text as follows.

The vertical distributions of concentrations and optical properties of carbonaceous aerosol particles, along with the impacts of vertical mixing, remain inadequately comprehended. This is primarily attributable to the dearth of in - situ observations of

meteorological and pollution conditions within the planetary boundary layer (PBL). The limited in - situ observations are predominantly concentrated on flat terrains, whereas they are scarce in complex terrains where intricate interactions among multi - scale circulations (such as gravity waves, mountain - plain winds, and mountain - valley breezes) occur. Consequently, the First Campaign on Boundary - Layer Meteorology and Pollution at Sichuan Basin (1st BLMP - SCB) was held from December 2018 to January 2019 to comprehensively comprehend the interactions between meteorology and pollution in the complex terrain. The vertical profiles of temperature, RH, and air pollutants, including CO, NO, $NO_2$, $O_3$, TVOCs, BC, and BrC, were monitored at three - hour intervals using instruments mounted on a tethered balloon. A Doppler Wind Lidar (Windcube 200s, Leosphere, France) was employed to acquire the profiles of winds, encompassing wind speed, wind direction, and vertical wind velocity. Drawing on the data from the 1st BLMP - SCB, this research examined the influence of mechanical turbulence and wind shear on the vertical profiles of air pollutants.

To more comprehensively elucidate the formation mechanism of primary and secondary carbonaceous aerosols, brown carbon (BrC) was further partitioned into primary sources ($BrC_{pri}$) and secondary formation ($BrC_{sec}$) by utilizing the minimum correlation coefficient between BrC and BC. The profiles of $BrC_{pri}$ and $BrC_{sec}$ can be acquired through this method. The concentrations of carbonaceous aerosols originating from primary sources (BC, $BrC_{pri}$) exhibit a rapid decline with the increase in altitude. In contrast, those resulting from secondary formation ($BrC_{sec}$) decrease at a slower rate and even reach high values at an altitude of 1.7–2.1 km ASL. Within the diurnal cycle, the $BrC_{sec}$ reaches a peak at an altitude of 1.4 km ASL at 11:00. Subsequently, it gradually descends towards the ground by 20:00, in conjunction with the development of the PBL turbulence. This phenomenon is primarily expounded upon from the perspectives of the downward intrusion of $BrC_{sec}$ in the upper atmosphere attributable to intense elevated turbulence.

To more comprehensively disclose the disparities between the vertical profiles of primary and secondary carbonaceous aerosols, the k - means clustering algorithm was utilized to classify the $BrC_{sec}$ profiles into three clusters, specifically: 1) relatively uniform vertical distributions; 2) high values at an altitude of 1.4 km ASL; and 3) more rapid decline with the increase in altitudes. Integrating the clustering analysis results with case studies, it is discovered that thermodynamic processes (temperature) serve as the dominant factors for the nighttime uniform profiles of $BrC_{sec}$. Nevertheless, during the daytime, $BrC_{sec}$ in the upper atmosphere can intrude downward into the PBL through strong dynamic processes (mechanical turbulence induced by wind shear), leading to high concentrations of $BrC_{sec}$ at 1.4 km ASL. The influences of long - range and regional transport were also investigated through backward trajectories and pollution rose plots. There is a minor disparity between BC and $BrC_{sec}$, and therefore, the discrepancy in the vertical structure between BC and $BrC_{sec}$ cannot be ascribed to long - range and regional transport. This research holds considerable significance for comprehensively comprehending the formation mechanism of the distinctive profile of secondary air pollutants and broadening the understanding of the interactions between meteorology and pollution in complex terrains.

Some interesting findings were acquired; however, this study is subject to certain limitations. The field campaign of the 1st BLMP - SCB was carried out solely at a rural location in the eastern foothills of the Tibetan Plateau (TP), and the observation duration was insufficiently long to yield more reliable conclusions. Consequently, the Second Boundary - Layer Meteorology and Pollution at SiChuan Basin (2nd BLMP - SCB) will be carried out at a rural location in Yaan City, which is situated more to the south than the site of the previous campaign. By integrating the data from the two campaigns, we aim to derive more general laws regarding the interactions between meteorology and pollution within the PBL in complex terrains, particularly in the sloped terrain transitioning from the SCB to the TP. The general conclusions hold significant importance in comprehending the formation mechanism of severe air

pollution in these complex topographical environments.

**Minor Issues**

1. Abstract: - I've rewritten the abstract to give the authors some guidelines on what kind of grammar mistakes need correcting. Please note that I have simply rewritten the abstract as it stands and the authors should not just use my example directly. I do not think this is a good abstract for an article, this seems more like a vague abstract you would see for a conference presentation. I strongly recommend the authors incorporate their actual results into the abstract in a more direct way.

**Response:** Thank you for your example abstract, which is very useful for improving the abstract. Some actual results were more directly incorporated into the abstract in the revised manuscript. The revised abstract was given with the blue text as follows.

The comprehension of the influence of the planetary boundary layer (PBL) processes on the vertical profiles of air pollutants in complex terrains remains highly restricted. In this study, data from the First Planetary Boundary- Layer Meteorology and Pollution campaign in the western Sichuan Basin (1st BLMP - SCB), carried out from December 2018 to January 2019, are utilized. The focus of the campaign is to provide data on the impact of the elevated turbulence on the profiles of particulate matter (PM) pollutants. This study focuses on two types of PM: black carbon (BC), as well as brown carbon from primary sources ($BrC_{pri}$) and that formed through secondary processes ($BrC_{sec}$). The concentrations of BC and $BrC_{pri}$ demonstrate a rapid decline as altitude increases, whereas the vertical profile of $BrC_{sec}$ concentration is variable. The results of k - means clustering reveal three distinct types of vertical profiles of $BrC_{sec}$: (1) relatively uniform vertical distributions (accounting for 17.28% of all profiles), (2) higher values at an altitude of 1.4 km above sea level (ASL) (16.05% of all profiles), and (3) more rapid decreases with the increase in altitude (66.67% of all profiles). Further analysis demonstrated that the nocturnal concentration profiles of $BrC_{sec}$ exhibit greater uniformity, featuring a minor peak at altitudes exceeding 1.7 km ASL. These profiles are more significantly influenced by thermodynamic processes.

During the daytime, $BrC_{sec}$ is mixed downward into the PBL through dynamic processes, namely, the elevated mechanical turbulence induced by wind shear. Throughout the campaign, both BC and $BrC_{sec}$ exhibit comparable regional and long-range transport characteristics. This study emphasizes the significance of the elevated turbulence in shaping the vertical profiles of PM pollutants in complex terrains. Specifically, the results are helpful for understanding formation mechanism of heavy air pollution within these complex topographic environments.

2. Line 32: "…filed campaign…" – should be "field." There are also several other places where "field" is misspelled and general spelling and grammar issues.

**Response:** Thank you for pointing out those errors. In the revised manuscript, the misspelling of "filed", as well as general spelling and grammar issues, were rectified comprehensively.

3. Line 61: "Aerosol-planetary boundary layer (PBL)…" – While I understand what the authors are saying here it can be a bit confusing. For clarity, I recommend the authors write something along the lines of: "The interactions between aerosols and the planetary boundary layer (PBL)…

**Response:** I express my gratitude for your reminder. The introduction section has been revised in accordance with your previous suggestions. The revised introduction places emphasis on the vertical profiles of carbonaceous aerosols and the corresponding formation mechanisms, especially the function of vertical mixing in complex terrain.

4. Line 68: "BC can also…" – typically starting a sentence with an acronym is to be avoided for readability purposes. This comment applies to many places in the article.

**Response:** I express my gratitude for your reminder. In the revised edition of our manuscript, sentences starting with an acronym were to be avoided to enhance readability.

5. Line 79: "…within PBL are…" – "…within the PBL are…" – this comment applies to all cases where PBL is used, "the" is necessary for readability purposes. Also applies to other cases like "Tibetan Plateau"

**Response:** Thank you for your reminder. We double-checked the manuscript and "the" will be added for the cases like "Tibetan Plateau" or "PBL" in the revised version of our manuscript.

6. Line 166-174: "A portable GPS…given by the manufacturer" – Very hard to follow section of text. It just all over the place.

**Response:** I sincerely appreciate your identification of that issue. I deeply apologize for the resulting confusion. The sentences spanning from Lines 166 to 174 have been revised to enhance comprehensibility, and similar instances throughout the manuscript have also been rectified.

An iMet-1 AB radiosonde, carried by the tethered balloon, was utilized to conduct the monitoring of the vertical profiles of temperature and RH. The measurement uncertainties of temperature and RH were ±0.3 °C and ±5%, respectively (Hosom et al., 1995). The radiosonde has been extensively utilized and verified (Haman et al., 2012), and there exists a negligible disparity compared to other radiosondes, such as the Vaisala RS92 (Trapp et al., 2016). The iMet-1 AB radiosonde and the Vaisala RS92 radiosonde were attached to the same 200-g balloon and launched from the identical location during the daytime. The measurements obtained from the two radiosondes exhibited extremely minor disparities in temperature. Specifically, the median difference remained below 0.5 K across all altitudes below 200 hPa. The RH measured by the iMet radiosonde exhibited a marginal decrease within the boundary layer (median values approximately -2%) and a slight increase within the 500 - 300 hPa layer (median values approximately +2%). Nevertheless, overall, the disparities were negligible throughout the troposphere (Trapp et al., 2016). The vertical resolution of temperature and RH for each profile ranged from 1 to 3 m.

7. Line 184: "0.2 and 0.05 Hz" – earlier in the paragraph temporal resolutions were given in seconds and not hertz. For improved readability it would be good to stick to one or the other.

**Response:** Thank you for your reminder. The temporal resolutions were unified to seconds throughout the manuscript for the improvement of readability.

**Reference**

Castro L.M., Pio C.A., Harrison R.M., Smith D.J.T., 1999. Carbonaceous aerosol in urban and rural European atmospheres: estimation of secondary organic carbon concentrations. *Atmospheric Environment*, 33(17), 2771–2781, DOI: 10.1016/S1352-2310(98)00331-8.

Cheng Y.H., Lin M.H., 2013. Real-time performance of the microAeth® AE51 and the effects of aerosol loading on its measurement results at a traffic site. *Aerosol and Air Quality Research*, 13(6), 853–1863.

Hagler G.S., Yelverton T.L., Vedantham R., Hansen A.D., Turner J.R., 2011. Post-processing method to reduce noise while preserving high time resolution in Aethalometer real-time black carbon data. *Aerosol and Air Quality Research*, 11, 539–546.

Hallar A.G., Brown S.S., Crosman E., Barsanti K.C., Cappa C.D., Faloona I., Fast J., Holmes H.A., Horel J., Lin J., Middlebrook A., Mitchell L., Murphy J., Womack C.C., Aneja V., Baasandorj M., Bahreini R., Banta R., Bray C., Brewer A., Caulton D., de Gouw J., De Wekker S., Farmer D.K., Gaston C.J., Hoch S., Hopkins F., Karle N.N., Kelly J.T., Kelly K., Lareau N., Lu K.D., Mauldin R.L., Mallia D.V., Martin R., Mendoza D., Oldroyd H.J., Pichugina Y., Pratt K.A., Saide P., Silva P.J., Simpson W., Stephens B.B., Stutz J., Sullivan A., 2021. Coupled air quality and boundary-layer meteorology in Western US basins during winter: design and rationale for a comprehensive study. *Bulletin of the American Meteorological Society*, 102(10), E2012–E2033.

Haman C.L., Lefer B., Morris G.A., 2012. Seasonal variability in the diurnal evolution of the boundary layer in a near-coastal urban environment. *Journal of*

*Atmospheric and Oceanic Technology*, 29(5), 697–710.

Hoffer A., Gelencsér A., Guyon P., Kiss G., Schmid O., Frank G. P., Artaxo P., Andreae M.O., 2006. Optical properties of humic-like substances (HULIS) in biomass-burning aerosols. *Atmospheric Chemistry and Physics*, 6, 3563–3570.

Hosom D.S., Weller R.A., Payne R.E., Prada K.E., 1995. The IMET (Improved Meteorology) ship and buoy systems. *Journal of Atmospheric and Oceanic Technology*, 12, 527–540.

Hsu Y.K., Holsen T.M., Hopke P.K., 2003. Comparison of hybrid receptor models to locate PCB sources in Chicago. *Atmospheric Environment*, 37, 545–562.

Kirchstetter T.W., Novakov T., Hobbs P.V., 2004. Evidence that the spectral dependence of light absorption by aerosols is affected by organic carbon. *Journal of Geophysical Research*, 109, D21208, doi:10.1029/2004JD004999.

Lan C., Liu H., Li D., Katul G.G., Finn D., 2018. Distinct turbulence structures in stably stratified boundary layers with weak and strong surface shear. *Journal of Geophysical Research: Atmospheres*, 123, 7839–7854.

Martilli A., Clappier A., Rotach M.W., 2002. An urban surface exchange parameterisation for mesoscale models. *Boundary-Layer Meteorology*, 104(2), 261–304.

Millet D.B., Donahue N.M., Pandis S.N., Polidori A., Stanier C.O., Turpin B.J., Goldstein A.H., 2005. Atmospheric volatile organic compound measurements during the Pittsburgh Air Quality Study: Results, interpretation, and quantification of primary and secondary contributions. *Journal of Geophysical Research-Atmospheres*, 110(D7), D07S07, DOI: 10.1029/2004JD004601.

Pang X.B., Chen L., Shi K., Wu F., Chen J., Fang S., Wang J., Xu M., 2021. A lightweight low-cost and multipollutant sensor package for aerial observations of air pollutants in atmospheric boundary layer. *Science of the Total Environment*, 764, 142828.

Park S.S., Hansen A.D.A., Cho S.Y., 2010. Measurement of real time black carbon for investigating spot loading effects of Aethalometer data. *Atmospheric Environment*, 44, 1449–1455.

Song W., Zhang Y., 2024. Analysis of TKE revenue and Expenditure characteristics and flux at different heights of the underlying surface of the southern Sichuan Forest. *Open Journal of Natural Science*, 12(5), 981–989.

Sun J., Mahrt L., Banta R.M., Pichugina Y.L., 2012. Turbulence regimes and turbulence intermittency in the stable boundary layer during CASES-99. *Journal of the Atmospheric Sciences*, 69(1), 338–351.

Syakur M.A., Khotimah B.K., Rochman E.M.S., Satoto B.D., 2018. Integration K-means clustering method and Elbow method for identification of the best customer profile cluster. *2nd International Conference on Vocational Education and Electrical Engineering (ICVEE)*, 336, DOI: 10.1088/1757-899X/336/1/012017.

Trapp R.J., Stensrud D.J., Coniglio M.C., Schumacher R.S., Baldwin M.E., Waugh S., Conlee D.T., 2016. Mobile radiosonde deployments during the mesoscale predictability experiment (MPEX): rapid and adaptive sampling of upscale convective feedbacks. *Bulletin of the American Meteorological Society*, 97(3), 329–336.

Yue P., Zhang Q., Wang R., Li Y., Wang S., 2015. Turbulence intensity and turbulent kinetic energy parameters over a heterogeneous terrain of Loess Plateau. *Advances in Atmospheric Sciences*, 32, 1291–1302.

---

## Author Comment (AC2)

**Responses to the Comments of Reviewer #2**

Thank you for your constructive suggestions, which is very useful for improving the readability of the manuscript. We have carefully addressed all the comments and suggestions you raised and provided a point - by - point response to each comment as follows. The revised manuscript looks like a new submission. The queries posed by the reviewer are presented in black text, the corresponding responses are showed in red text, and the revised content in the manuscript is in blue text.

**General Comments**

1. Most of my comments pertain to the language presented in the manuscript. I understand that English is not everyone's first language, and I apologize in advance if my remarks come across as insensitive in any way. I strongly recommend that the authors have this manuscript proofread by someone proficient in English. Doing so will help elucidate some of the language used and improve the overall quality of the manuscript. Please note that some aspects of the research presented here are outside my area of expertise, so I apologize in advance if any of my comments seem basic.

**Response:** We express our sincere gratitude for your suggestions, which are of great significance for enhancing the quality and readability of the manuscript. The manuscript was meticulously proofread by individuals proficient in English, which contributed to clarifying certain language usage and elevating the overall quality of the manuscript.

2. Data and Methods: I believe this section needs substantial revision, particularly with regard to:

- Site and instrumentation description

- Data utilization and quality control, in particular:

  o Validation of deployed instrument measurements to ground based instruments

  o Quality control techniques

- Analysis techniques in sections detailing:

  o K-means cluster analysis

o Utilization of turbulence parameters

**Response:** We express our sincere gratitude for your constructive suggestions, which have proven highly valuable for the enhancement of our manuscript. In the revised version, the section titled "Data and Methods" has been comprehensively rewritten and reorganized, and it is now divided into the following sub - sections. The specific details can be found within the revised main body of the text.

2.1 Experimental setup and data collection

2.1.1 Experimental setup

2.1.2 Data collection and quality control

2.2 Methods

2.2.1 Identification of potential source regions for BC and BrC

2.2.2 K-means cluster analysis

2.2.3 Utilization of turbulence parameters

3. Results:

  - In the manuscript, some figures are referenced with variables that do not appear in the figures themselves. For example, the manuscript mentions a height reference to Figure S1, but the figure does not contain any height dimension.

  - I believe that some figures in the supplement should be included in the manuscript itself. For example, Figure S1 is referenced frequently in alongside Figure 3. It makes it easier to read and understand when both figures are present the manuscript rather than having to reference the supplement section for figures.

**Response:** We express our sincere gratitude for your constructive suggestions. These suggestions are of great significance for enhancing the readability of our manuscript. In accordance with your suggestions, all figures have been reorganized and redrawn, and variables that do not appear within the figures themselves have been added. In Figures 3, 4, and S3, the size of the squares or dots represents altitude, with the altitude ranging from 0.65 to 2.70 km as the size of the squares or dots increases. Figure S1 in the supplementary information has been relocated to the main text and renamed as Figure 3, and the former Figure 3 has been renamed as Figure 4.

[Figure]

Fig. 3 Diurnal variations in the relationships between the profiles of BC and $BrC_{pri}$ during the campaign. The correlation coefficients ($r$) exceed 0.99 for all the relationships. The coefficients of divergence (COD) were calculated using the following equation: $COD_{jk}=\sqrt{1/p\times\sum_{1}^{p}\left[\left(x_{ij}-x_{ik}\right)/\left(x_{ij}+x_{ik}\right)\right]^{2}}$ . The black solid lines represent the 1:1 line. The situations during daytime (08:00, 11:00, 14:00, and 17:00, local time) and nighttime (02:00, 05:00, 20:00, and 23:00) were delineated by transparent yellow and gray boxes, respectively. The size of the squares signifies altitude, with the altitude ranging from 0.65 to 2.70 km as the squares increase in size.

**Major comments**

1. Abstract (Lines 32 - 50): Abstract needs to be rewritten, the language makes it a bit hard to follow. There are grammatical and sentence structure issues that I believe after resolving, can really improve the flow of the abstract and elucidate the content for the reader.

**Response:** I express my gratitude for your suggestions. The abstract has been rephrased, and several key actual findings have been incorporated into the revised abstract. Moreover, the grammatical and sentence structure issues in the abstract have been addressed in the revision, which is presented in blue text as follows.

The comprehension of the influence of planetary boundary layer (PBL) processes on the vertical profiles of air pollutants in complex terrains remains highly restricted. In this study, data from the First Planetary Boundary- Layer Meteorology and Pollution campaign in the western Sichuan Basin (1st BLMP - SCB), carried out from December 2018 to January 2019, are utilized. The focus of the campaign is to provide data on the impact of the elevated turbulence on the profiles of particulate matter (PM) pollutants. This study focuses on two types of PM: black carbon (BC), as well as brown carbon from primary sources ($BrC_{pri}$) and that formed through secondary processes ($BrC_{sec}$). The concentrations of BC and $BrC_{pri}$ demonstrate a rapid decline as altitude increases, whereas the vertical profile of $BrC_{sec}$ concentration is variable. The results of k - means clustering reveal three distinct types of vertical profiles of $BrC_{sec}$: (1) relatively uniform vertical distributions (accounting for 17.28% of all profiles), (2) higher values at an altitude of 1.4 km above sea level (ASL) (16.05% of all profiles), and (3) more rapid decreases with the increase in altitude (66.67% of all profiles). Further analysis demonstrated that the nocturnal concentration profiles of $BrC_{sec}$ exhibit greater uniformity, featuring a minor peak at altitudes exceeding 1.7 km ASL. These profiles are more significantly influenced by thermodynamic processes. During the daytime, $BrC_{sec}$ is mixed downward into the PBL through dynamic processes, namely, the elevated mechanical turbulence induced by wind shear. Throughout the campaign, both BC and $BrC_{sec}$ exhibit comparable regional and long-range transport characteristics. This study emphasizes the significance of the elevated turbulence in shaping the vertical profiles of PM pollutants in complex terrains. Specifically, the results are helpful for understanding formation mechanism of heavy air pollution within these complex topographic environments.

2. All Figures in manuscript: Text on the figures are too small and hard to look at initially for information. For example: Figures 4, 5, S4, and S5 have embedded text that is critical for interpreting the figures, but the small text make this difficult. I recommend enlarging the text in all figures for better clarity.

**Response:** Thank you for your constructive suggestions. The embedded text within all figures has been enlarged to enhance clarity.

3. Lines 134 - 135: "The first field campaign of Boundary Layer Meteorology and Pollution at SiChuan Basin (BLMP-SCB) was conducted at a rural site (Sanbacun, 103°40′38″ E, 30°54′59″ N) of eastern foothills of Tibetan Plateau in winter of 2018, lasting about 40 days (Fig. 1)." - Please include elevation for those not familiar with the region.

**Response:** Thank you for identifying that issue. The statement "... at a rural site (Sanbacun, 103°40′38″ E, 30°54′59″ N) ..." has been amended to "... at a rural site (Sanbacun, 103°40′38″ E, 30°54′59″ N, 650 m) ...". The elevation of Sanbacun has been incorporated into the revised version of our manuscript.

4. Lines 146 - 147: "The performances of the sensors were verified by comparing with on-ground reference instruments (Pang et al., 2021) …" - What instruments specifically? Can you provide some statistics describing these differences? (For example: R2)?

**Response:** I sincerely appreciate your reminder. The sentences in Lines 146 - 147 have been revised in accordance with your suggestions. The revised contents are presented in blue text as follows.

The performance of the package (gaseous and particulate matter sensors) was validated through comparison with on - ground reference equipment (Model 49C, Model 42i, Model 450i, Model 48i - TLE, Model 5030i, Thermal Fisher, USA) with coefficients of determination ($R^2$) ranging from 0.81 to 0.93 (Pang et al., 2021).

5. Lines 168 - 170: "The radiosonde has been widely used and validated (Haman et al., 2012), and there is a very slight difference with the other radiosondes such as Vaisala RS92 (Trapp et al., 2016)." – Validated how? Again, I think providing quantitative statistics that describe the difference would be beneficial.

**Response:** Thank you for your suggestions. The sentences in Lines 168 - 170 have been revised in accordance with your suggestions. The revised contents are presented in blue text as follows.

The radiosonde has been extensively utilized and verified (Haman et al., 2012), and there exists a negligible disparity compared to other radiosondes, such as the Vaisala RS92 (Trapp et al., 2016). The iMet-1 AB radiosonde and the Vaisala RS92 radiosonde were attached to the same 200-g balloon and launched from the identical location during the daytime. The measurements obtained from the two radiosondes exhibited extremely minor disparities in temperature. Specifically, the median difference remained below 0.5 K across all altitudes below 200 hPa. The RH measured by the iMet radiosonde exhibited a marginal decrease within the boundary layer (median values approximately -2%) and a slight increase within the 500 - 300 hPa layer (median values approximately +2%). Nevertheless, overall, the disparities were negligible throughout the troposphere (Trapp et al., 2016).

6. Lines 217 – 219: "Clustering analysis was used to divide the UVPM profiles during the campaign into three groups with comparable vertical structure of UVPM within groups." – What are the groups? Explain in greater detail what each group signifies. How did you decide that three clusters were appropriate for this analysis? I suggest providing some sort of clustering validation to determine the correct number of clusters, such as "The Elbow Method" or even "Silhouette score" could be beneficial. I recommend the authors look into this and provide context, you do not necessarily need to provide plots, but I think a simple reference would suffice.

**Response:** We express our sincere gratitude for your constructive suggestions. These suggestions are of great significance in enhancing the readability of our manuscript. The characteristics of each cluster were presented in a table, and the number of clusters was determined using the Elbow Method. The relevant reference was provided in the revised manuscript. The relevant contents are presented in blue text as follows.

The determination of the number of clusters is crucial for attaining novel and accurate understandings. Drawing upon the Elbow Method (Syakur et al., 2018) and the characteristic profiles of air pollutants, k - means clustering analysis was employed to partition the $BrC_{sec}$ profiles during the campaign into three clusters, with each cluster featuring a comparable vertical structure of $BrC_{sec}$. The characteristics of each cluster of the $BrC_{sec}$ profile are presented in Table 2.

Table 2 Characteristics of the three clusters of $BrC_{sec}$ profiles.

| Cluster | Descriptions | Frequency |
|---------|-------------|-----------|
| 1 | Relatively uniform vertical distributions | 17.28% |
| 2 | Higher values at an altitude of 1.4 km above sea level (ASL) | 16.05% |
| 3 | More rapid decreases with the increase in altitude | 66.67% |

7. Lines 231 – 233: "The ONA algorithm results in significant noise reductions and much more reasonable temporal changes in mass concentrations of carbonaceous particles (Cheng and Lin 2013; Park et al., 2010)." – What exactly is the "ONA algorithm"? I think a more in-depth description into the mechanics and procedures of this algorithm is necessary. What percentage of data required the ONA technique? I think including this is also necessary to include in the manuscript for context.

**Response:** Thank you for your constructive suggestions. As you suggested, the highly simplistic depiction of the ONA algorithm impeded the comprehension of the technique. The revised explications are presented in blue text as follows.

The micorAeth MA200 might yield negative values under conditions of lower mass concentrations and higher temporal resolution, which can account for up to 30% of the uncertainty associated with the filter - based optical attenuation technique (Hagler et al., 2011). Consequently, the raw data acquired for vertical profiles must be

rectified prior to analyzing their characteristics, particularly for in - situ observations at high altitudes. The mere removal of negative values is an inappropriate approach, as it would neglect the corresponding positive fluctuations caused by noise and lead to an upward bias in the final data. Averaging data over an extended time frame typically mitigates the noise within the signal; however, this may conflict with the requirement for high temporal - resolution data. Post - processing strategies such as moving averages or advanced mathematical techniques can be utilized to isolate the noise and reconstruct the time series (Kostelich and Schreiber, 1993). Nevertheless, these methods fail to leverage the ATN values associated with the internal load rate of the filter and the knowledge regarding the successive difference characteristic of the MA200. The optical noise - reduction averaging (ONA) algorithm, devised by Hagler et al. (2011), aims to perform adaptive time - averaging of carbonaceous components so as to mitigate the noise in BC data.

The ONA algorithm conducts smoothing processing on the time series of carbonaceous particles via a user - specified minimum attenuation change ($\Delta ATN_{min}$). For a given concentration of carbonaceous aerosols, this process leads to an adjusted timebase ($\Delta t'$). When the concentration reaches a sufficiently high level or the intrinsic timebase is long, $\Delta ATN$ will exceed $\Delta ATN_{min}$, and the intrinsic time resolution will be maintained. Nevertheless, in the case of relatively low concentrations or short timebases, $\Delta ATN$ will be lower than $\Delta ATN_{min}$, and the time series will be smoothed over the time intervals $\Delta t' > \Delta t$ required to attain $\Delta ATN_{min}$. A second constraint is that the ATN value at the conclusion of the interval $\Delta t'$ must be the final instance of that value within the remaining part of the time - series for that specific sample spot. Consequently, $\Delta t_i'$ is extended to the ultimate occurrence of that ATN value. The frequency of negative values is reduced by applying the constraint. This is because when the ATN value returns to the same level later in the time - series, it implies that $\Delta ATN < 0$ at that specific time step, which leads to a negative concentration. A consequence of the second constraint is that a certain level of smoothing exists even when $\Delta ATN_{min}$ equals zero. In principle, the average

concentration of carbonaceous particles during the time interval $\Delta t_i$' can be calculated using $\Delta ATN_i/\Delta t_i$'. Nevertheless, the duration of light transmission measurement at a high temporal resolution is relatively brief, rendering it prone to noise interference. The mean concentration of carbonaceous particles during the time interval $\Delta t_i$' is calculated by averaging the set of concentrations reported at the intrinsic timebase within that interval. The incidence of negative concentrations should be less than 30% of the datasets to guarantee data quality. This is a straightforward method for addressing the noise in real - time data from the micorAeth MA200. It achieves this by dynamically adjusting the competing factors of averaging time and noise, thereby preserving the highest possible temporal resolution within the datasets. The algorithm leads to substantial reductions in noise and considerably more rational temporal variations in the mass concentrations of carbonaceous particles (Cheng and Lin, 2013; Park et al., 2010). The program was employed to conduct post - processing on the negative values obtained from our real - time profile measurements.

8.  Line 237: The authors mention UVPMpri, but do not reference what it is or what it means? I assume this means organic carbon from primary sources, but this is not directly stated in the prior sections.

**Response:** Thank you for your reminder. In fact, the ultraviolet particulate matter (UVPM) corresponds to brown carbon (BrC). Consequently, UVPM has been uniformly substituted with BrC across the entirety of the manuscript. Brown carbon stemming from primary sources and secondary formation has been respectively designated as $BrC_{pri}$ and $BrC_{sec}$. These designations have been explicitly stated in the preceding sections of the revised manuscript.

9.  Lines 252 – 254: At this stage in the manuscript, the distinction between UVPMpri and UVPMsec in Equations 1 and 2 is not clear to me. In Eq. 2 UVPM is referenced, but the manuscript does not specify how UVPM differs from UVPMpri and UVPMsec.

**Response:** I sincerely appreciate your attention to this matter. I deeply apologize for

the confusion arising from the ambiguous expression. In the revised manuscript, UVPM, UVPMpri, and UVPMsec have been respectively substituted with BrC, BrCpri, and BrCsec, and their definitions are provided upon their initial appearance. The revised text is presented in blue as follows.

The estimation of secondary brown carbon ($BrC_{sec}$) holds significant importance in ascertaining the proportion of $BrC_{sec}$ within BrC. Initially, the minimum ratio of BrC to BC, denoted as $(BrC/BC)_{min}$, was employed as a surrogate for the ratio of $BrC_{pri}$ to BrC ($BrC_{pri}/BrC$) to estimate the mass concentrations of $BrC_{sec}$ (Castro et al., 1999). Nevertheless, numerous studies have indicated that the $(BrC/BC)_{min}$ demonstrates a certain level of randomness in actual observations, which results in substantial errors, particularly for the low BC concentrations in high - altitude regions. To tackle this issue, Lim and Turpin (2002) put forward the approach of arranging the BrC/BC ratios in ascending order and substituting the $BrC_{pri}/BC$ ratio with the mean value of the top 10%–20% of the data. However, there is a dearth of a universally applicable criterion for determining the appropriate percentile range. In light of the disparate sources of $BrC_{sec}$ and BC, Millet et al. (2005) put forward a method for estimating $BrC_{sec}$ concentrations by utilizing the minimum correlation coefficient between BrC and BC. This methodology aims to determine the $BrC_{pri}/BC$ ratio (designated as $(BrC/BC)_{pri}$) at which the correlation between $BrC_{sec}$ and BC reaches its minimum, and this ratio is employed as the $BrC_{pri}/BC$ ratio. Adopting this approach, Wu and Yu (2016) devised a toolkit within Igor Pro for calculating the mass concentration of $BrC_{sec}$. This development notably improved the precision of $BrC_{sec}$ estimation, as presented in Eqs. (1) and (2).

$$BrC_{pri} = (BrC/BC)_{pri} \times BC, \qquad (1)$$

$$BrC_{sec} = BrC - BrC_{pri}. \qquad (2)$$

In Eq. (1), $(BrC/BC)_{pri}$ denotes the ratio of the concentrations of $BrC_{pri}$ to BC during the campaign. Based on the measurements of BrC and BC, $BrC_{pri}$ and $BrC_{sec}$ can be estimated by means of Eqs. (1) and (2).

10. Section 2.5 Calculation of mechanical turbulence and wind shear: It seems like the authors want to isolate shear as a main contributor to atmospheric exchange and transport. However, it is not clear to me how this is directly achieved. The authors mention the TKE equation and shear terms, but I recommend the authors look into the TKE Budget equation, if possible, given the instrumentation deployed and measurements available. The TKE budget equation provides more detailed insight into the generation, transport, and dissipation of turbulence across a specific transect.

**Response:** I express my sincere gratitude for your constructive suggestions, which have notably enhanced the readability of the manuscript. In accordance with the research findings of Lan et al. (2018) and Sun et al. (2012), the term "mechanical turbulence index" has been modified to "turbulent velocity scale" in the revised manuscript. As you suggested, The TKE budget equation offers a more comprehensive and in - depth understanding of the generation, transport, and dissipation of turbulence along a specific transect. In the context of unstable stratification, a significant correlation exists between TKE and the effects of mechanical shear and buoyancy (Yue et al., 2015). During the campaign, the buoyancy effect is notably suppressed on cloudy days (Song and Zhang, 2024). Furthermore, considering the constraints of the deployed instrumentation and the available measurements with relatively low temporal resolution, this study only analyzed the mechanical shear (including wind speed shear and directional shear). The revised contents were showed with the blue text as follows.

Turbulent kinetic energy (TKE) serves as an indicator of turbulence intensity, and it represents a crucial variable for the comprehension of the exchanges of energy, water vapor, and greenhouse gases between the land surface and the atmosphere. The turbulent velocity scale, defined as $V_{TKE} = \left[ (1/2)\left( \overline{u'^2} + \overline{v'^2} + \overline{w'^2} \right) \right]^{1/2} = \sqrt{TKE}$, has been widely employed to characterize turbulence intensity (Lan et al., 2018; Sun et al., 2012). Here, $u$, $v$, and $w$ denote the zonal, meridional, and vertical wind components

respectively, while $u'$, $v'$, and $w'$ signify the standard deviation of each variable. The vertical profiles of $V_{TKE}$ can be derived from the aforementioned equation. A positive correlation exists between the $V_{TKE}$ and mechanical turbulence, such that an increase in $V_{TKE}$ corresponds to an enhancement of mechanical turbulence. The combination of the profiles of $V_{TKE}$ and air pollutants can be employed to gain a more comprehensive understanding of the downward transport of air pollutants at the eastern foothills of the Tibetan Plateau.

The TKE budget equation offers a more comprehensive and in - depth understanding of the generation, transport, and dissipation of turbulence along a specific transect (Martilli et al., 2002, Eq. 4).

$$\frac{1}{2}\frac{\partial e^2}{\partial t} = -\overline{u'w'}\frac{\partial \overline{u}}{\partial z} + \frac{g}{T}\left(\overline{w'\theta'}\right) - \frac{1}{2\rho}\frac{\partial \overline{\left(w'e^2\right)}}{\partial z} - \frac{1}{\rho}\frac{\partial \overline{w'p'}}{\partial z} - v\left(\overline{\frac{\partial u_i'}{\partial x_j}\frac{\partial u_i'}{\partial x_j}}\right), \qquad (4)$$

where $e$ represents the TKE, $t$ denotes time, $g$ signifies the acceleration due to gravity, $T$ stands for temperature, $\theta$ represents the potential temperature, $\overline{w'\theta'}$ refers to the turbulent heat flux, $\overline{u'w'}$ indicates the turbulent momentum, $\rho$ denotes the air density, $u$ represents the wind speed, $z$ represents the observational height, $w'$ represents the vertical wind velocity fluctuation, and $p'$ represents the atmospheric pressure pulsation. The value of TKE is primarily influenced by mechanical shear and buoyancy effects (represented by the first two terms on the right - hand side of Equation 4). In the context of unstable stratification, a significant correlation exists between TKE and the effects of mechanical shear and buoyancy (Yue et al., 2015). During the campaign, the buoyancy effect is notably suppressed on cloudy days (Song and Zhang, 2024). Furthermore, considering the constraints of the deployed instrumentation and the available measurements with relatively low temporal resolution, this study only analyzed the mechanical shear (including wind speed shear and directional shear).

11. Lines 310 – 312: "In order to better understand the mechanisms of the more

uniform profiles of UVPMsec as compared to those of BC and UVPMpri, we firstly analyzed the relationships between UVPMsec or UVPMpri and BC (Figs. 3 and S1)." - This requires a bit of rephrasing to understand. The authors reference Figures 3 and S1 frequently in this section. I suggest moving Figure S1 into the main text. This just makes it easier to reference especially in a section of the results where both figures are needed together to clarify the interpretation.

**Response:** Thank you for your suggestions. In the revised manuscript, Figure S1 has been relocated to the main text and renamed as Fig. 3, whereas the former Fig. 3 has been renamed as Fig. 4. These two figures have been reorganized and redrawn to distinguish the situations between daytime and nighttime as well as different altitudes.

12. Lines 323 – 326: "Specifically, the differences between BC and UVPMpri are getting smaller and smaller with the increasing altitudes at 02:00–11:00 and 23:00, while those are independent on altitudes with the low COD values (0.039–0.098) at 14:00–20:00 (Fig. S1)." – Authors mention altitude, but altitude is not on Figure S1.

**Response:** Thank you for your reminder. Figure S1 has been reorganized and redrawn to differentiate the scenarios between daytime and nighttime, along with those at different altitudes. The revised figure is presented in the aforementioned questions.

13. Lines 329 – 330: "During the daytime, UVPMsec firstly increased with BC concentrations and then decreased gradually as the increased BC." – Rephrase, this is a bit confusing for the reader to understand.

**Response:** Thank you for your suggestions. The sentence in Lines 329-330 will be rephrased to "During the daytime, the $BrC_{sec}$ exhibited an increasing trend with the rise in BC concentrations. Subsequently, as the BC concentrations continued to increase, the $BrC_{sec}$ less varied." in the revised version of our manuscript.

14. Lines 493 – 496: ". Furthermore, the UVPMsec peaks well correspond to the strong descending motion and wind shear, and thus the UVPMsec peaks at the upper air on 7 January 2019 are mainly modulated by dynamic processes instead of

thermodynamic processes." – Not sure what "peak" is referenced here, not clear by looking at the figure(s).

**Response:** Thank you for your reminder. The figure has been modified to a time - height - concentration plot to facilitate the more straightforward discernment of the concurrent variations between the UVPMsec peaks, vertical motion, and wind shear. The revised Figure 8 is presented as follows.

[Figure]

Fig. 8 Diurnal variations in vertical profiles of air pollutants (BrC$_{sec}$, BC, BrC$_{sec}$/BrC ratio, and PM$_1$) and meteorological factors (temperature, $w$, Sh, $S_{vec}$, and $S_{dir}$) on 7 January 2019. The black arrows indicate the temporal movement of the peaks of profiles of BrC$_{sec}$, Sh, and $S_{vec}$.

15. Lines 502 – 503: "As the surface is heated up and PBL developed during the daytime…" – I would suggest adding a plot that shows a time series of PBL development with height either from the Lidar or other remote sensing instruments (if possible). This helps visualize how deep and fast the PBL grows throughout the day.

**Response:** Thank you for your suggestions. The horizontal wind speed and direction,

as well as the vertical wind velocity, were observed on an hourly basis by a Doppler Wind Lidar at Sanbacun. The development of the PBL can be visualized through the diurnal variations of the vertical profiles of wind vectors within the x–y plane on 1 January 2019 (upper panel of Fig. S8). Moreover, the PBL height was calculated by means of the potential temperature gradient method using radiosonde data acquired from the tethered balloon, which is presented in the lower panel of Fig. S6. As can be discerned from the figure, the PBL developed rapidly after the early - morning hours (05:00), reached its maximum depth after noon (14:00), and subsequently declined gradually.

[Figure]

Fig. S6 Diurnal variations of the hourly vertical profiles of wind vectors within the x–y plane (upper panel) and the 3-h PBL height (lower panel) on 1 January 2019. The colors assigned to the wind vectors denote the vertical wind velocity ($w$). The PBL height was calculated via the potential temperature gradient method using radiosonde data obtained from the tethered balloon ($\frac{d\theta}{dZ}|_{Z=h_i}=0$).

**Minor Comments:**

1. All figures: Figures that reference time (For example: Fig 3), are these all in local time? UTC?

**Response:** Thank you for your reminder. Throughout the manuscript, the local time (Beijing Time = UTC + 8) was utilized. This statement will be presented in the "2 Data and methods" section of the revised version of our manuscript.

2. Line 130: "… understanding the change in air pollutants and then taking targeted measures." – What do you mean by "taking targeted measures"?

**Response:** I express my gratitude for your attention to this matter. The introduction section has been rewritten and reorganized in accordance with the theme of this study. Moreover, the statement "taking targeted measures" has been removed from the revised manuscript.

3. Lines 173 – 174: "The uncertainty of temperature and RH measurements was ±0.3 °C and ±5% given by the manufacturer." – Provide citation of manufacturer.

**Response:** Thank you for your valuable suggestion. The sentence spanning Lines 173 - 174 has been revised to "The measurement uncertainties of temperature and relative humidity (RH) were ±0.3 °C and ±5%, respectively (Hosom et al., 1995)". The relevant reference has been cited in the revised edition of our manuscript.

4. Line 183: "The DBS mode was used in this campaign." – What is "DBS mode"? I suggest the authors describe this in greater detail. Provide citations.

**Response:** I sincerely appreciate your constructive suggestions, which are of great significance for enhancing the readability of the manuscript. During the campaign of the 1st BLMP-SCB, the Doppler beam swinging (DBS) mode was employed to present profiles of wind speed and direction. By shifting the beam among a series of four radial wind directions, which are generally at an elevation of approximately 60° and mutually perpendicular, the Doppler shift and consequently the line - of - sight

(LOS) velocity can be calculated for the DBS mode (Lundquist et al., 2015). Explanations will be incorporated into the revised manuscript.

5. Lines 189 – 191: What instruments were on the tower? Can you provide model #'s, also basic statistics on comparison between Lidar and sonic wind anemometer values?

**Response:** Thank you for your inquiries and recommendations. The revised content is presented in blue text as follows.

The wind data measured by the Lidar were verified through comparison with the on - site wind observations obtained by a sonic wind anemometer (Windmaster, Gill Instruments, Lymington, Hampshire, UK) installed on the Beijing 325 - m meteorological tower. The horizontal wind speed measurements obtained from the Doppler Wind Lidar exhibited a high degree of consistency with those acquired by the sonic wind anemometer, as evidenced by high coefficients of determination ($R^2$: 0.90–0.97). Furthermore, the results indicated a high degree of consistency between the two measurements of the dominant wind direction (Dai et al., 2020).

6. Lines 298 – 300: "Unlike UVPMpri profiles, the vertical distributions of secondary UVPM (UVPMsec) were more uniform, and the differences among the profiles were more significant than UVPMpri profiles." – This is a bit unclear and should be rephrased for clarity.

**Response:** Thanks for your reminder. In the revised manuscript, the sentence spanning Lines 298 - 300 will be rephrased as follows for the sake of clarity: "In comparison with the profiles of $BrC_{pri}$, the profiles of $BrC_{sec}$ exhibited greater uniformity, and the disparities among the $BrC_{sec}$ profiles were more pronounced than those among the $BrC_{pri}$ profiles at different time points."

7. Lines 346 – 348: "Fig. S3 showed the relationships between UVPMsec/UVPM ratio and UVPMpri or UVPMsec concentrations at the varying altitude ranges at the different times of the day." – Rephrase for clarity. What do you mean by "UVPMpri

or UVPMsec"? Do you mean: "Figure S3 showed the relationships between the UVPMsec/UVPM ratio and UVPMpri, as well as between the UVPMsec/UVPM ratio and UVPMsec."?

**Response:** Thank you for your suggestions. In the revised manuscript, the sentence in Lines 346-348 will be rephrased to "Figure S4 depicts the relationships between the ratio of $BrC_{sec}$ to BrC and $BrC_{pri}$, as well as between the ratio of $BrC_{sec}$ to BrC and $BrC_{sec}$ concentrations across different altitude ranges at various time points throughout the day." for clarity.

**Reference**

Castro L.M., Pio C.A., Harrison R.M., Smith D.J.T., 1999. Carbonaceous aerosol in urban and rural European atmospheres: estimation of secondary organic carbon concentrations. *Atmospheric Environment*, 33(17), 2771–2781, DOI: 10.1016/S1352-2310(98)00331-8.

Cheng Y.H., Lin M.H., 2013. Real-time performance of the microAeth® AE51 and the effects of aerosol loading on its measurement results at a traffic site. *Aerosol and Air Quality Research*, 13(6), 853–1863.

Dai L.D., Coauthors, 2020. Multilevel validation of Doppler Wind Lidar by the 325 m meteorological tower in the planetary boundary layer of Beijing. *Atmosphere*, 11, 1051, doi:10.3390/atmos11101051.

Hagler G.S., Yelverton T.L., Vedantham R., Hansen A.D., Turner J.R., 2011. Post-processing method to reduce noise while preserving high time resolution in Aethalometer real-time black carbon data. *Aerosol and Air Quality Research*, 11, 539–546.

Haman C.L., Lefer B., Morris G.A., 2012. Seasonal variability in the diurnal evolution of the boundary layer in a near-coastal urban environment. *Journal of Atmospheric and Oceanic Technology*, 29(5), 697–710.

Hosom D.S., Weller R.A., Payne R.E., Prada K.E., 1995. The IMET (Improved Meteorology) ship and buoy systems. *Journal of Atmospheric and Oceanic Technology*, 12, 527–540.

Lan C., Liu H., Li D., Katul G.G., Finn D., 2018. Distinct turbulence structures in stably stratified boundary layers with weak and strong surface shear. *Journal of Geophysical Research: Atmospheres*, 123, 7839–7854.

Lundquist J.K., Churchfield M.J., Lee S., Clifton A., 2015. Quantifying error of lidar and sodar Doppler beam swinging measurements of wind turbine wakes using computational fluid dynamics. *Atmospheric Measurement Techniques*, 8, 907–920.

Martilli A., Clappier A., Rotach M.W., 2002. An urban surface exchange parameterisation for mesoscale models. *Boundary-Layer Meteorology*, 104(2), 261–304.

Millet D.B., Donahue N.M., Pandis S.N., Polidori A., Stanier C.O., Turpin B.J., Goldstein A.H., 2005. Atmospheric volatile organic compound measurements during the Pittsburgh Air Quality Study: Results, interpretation, and quantification of primary and secondary contributions. *Journal of Geophysical Research-Atmospheres*, 110(D7), D07S07, DOI: 10.1029/2004JD004601.

Pang X.B., Chen L., Shi K., Wu F., Chen J., Fang S., Wang J., Xu M., 2021. A lightweight low-cost and multipollutant sensor package for aerial observations of air pollutants in atmospheric boundary layer. *Science of the Total Environment*, 764, 142828.

Park S.S., Hansen A.D.A., Cho S.Y., 2010. Measurement of real time black carbon for investigating spot loading effects of Aethalometer data. *Atmospheric Environment*, 44, 1449–1455.

Sun J., Mahrt L., Banta R.M., Pichugina Y.L., 2012. Turbulence regimes and turbulence intermittency in the stable boundary layer during CASES-99. *Journal of the Atmospheric Sciences*, 69(1), 338–351.

Syakur M.A., Khotimah B.K., Rochman E.M.S., Satoto B.D., 2018. Integration K-means clustering method and Elbow method for identification of the best customer profile cluster. *2nd International Conference on Vocational Education and Electrical Engineering (ICVEE)*, 336, DOI: 10.1088/1757-899X/336/1/012017.

Trapp R.J., Stensrud D.J., Coniglio M.C., Schumacher R.S., Baldwin M.E., Waugh S., Conlee D.T., 2016. Mobile radiosonde deployments during the mesoscale

predictability experiment (MPEX): rapid and adaptive sampling of upscale convective feedbacks. *Bulletin of the American Meteorological Society*, 97(3), 329–336.

Yue P., Zhang Q., Wang R., Li Y., Wang S., 2015. Turbulence intensity and turbulent kinetic energy parameters over a heterogeneous terrain of Loess Plateau. *Advances in Atmospheric Sciences*, 32, 1291–1302.